# Dandelion pappus morphing is actuated by radially patterned material swelling

Madeleine Seale [1,2,3], Annamaria Kiss [4], Simone Bovio [4], Ignazio Maria Viola [2], Enrico Mastropaolo[2], Arezki Boudaoud [4,5✉] & Naomi Nakayama [1,6✉]

Plants generate motion by absorbing and releasing water. Many Asteraceae plants, such as the dandelion, have a hairy pappus that can close depending on moisture levels to modify dispersal. Here we demonstrate the relationship between structure and function of the underlying hygroscopic actuator. By investigating the structure and properties of the actuator cell walls, we identify the mechanism by which the dandelion pappus closes. We developed a structural computational model that can capture observed pappus closing and used it to explore the critical design features. We find that the actuator relies on the radial arrangement of vascular bundles and surrounding tissues around a central cavity. This allows heterogeneous swelling in a radially symmetric manner to co-ordinate movements of the hairs attached at the upper flank. This actuator is a derivative of bilayer structures, which is radial and can synchronise the movement of a planar or lateral attachment. The simple, material-based mechanism presents a promising biomimetic potential in robotics and functional materials.

[1] School of Biological Sciences, University of Edinburgh, Edinburgh EH9 3BF, UK. [2] School of Engineering, University of Edinburgh, Edinburgh EH9 3FF, UK. [3] Department of Plant Sciences, University of Oxford, South Parks Road, Oxford OX1 3RB, UK. [4] RDP, ENS de Lyon, Université de Lyon, UCB Lyon 1, INRAE, CNRS, 69364, Lyon, Cedex 07, France. [5] LadHyX, CNRS, Ecole Polytechnique, Institut Polytechnique de Paris, 91128, Palaiseau Cedex, France. [6] Department of Bioengineering, Imperial College, South Kensington, London SW7 2AZ, UK. ✉email: arezki.boudaoud@polytechnique.edu; n.nakayama@imperial.ac.uk

Movement of body parts are typically mediated by specialised hinge structures - actuators - in biological and engineered systems. Biological actuators consist of continuous structures, and differential expansion within the actuator drives reversible movement and morphing. A thematic example is bilayer structure, in which two sides of a planar or cylindrical body expand or shrink more to cause bending or twisting. Inspired by the plethora of examples from nature, diverse designs have been developed for bilayer soft robotic actuators. Hygroscopic plant movements have been particularly relevant for biomimetic engineering and design as some of them do not rely on inherently biologically active processes[1]. Instead, they can highlight structural features that have been tuned by evolution to optimise mechanical efficiency or use of materials.

Plant movements and morphing are generally driven by changes in hydration[2]. This can be actively regulated by altering solute concentrations to manipulate osmotic gradients or by increasing water uptake and the prevalence of aquaporins[2,3]. Active water movement occurs in the opening and closing of stomata and the leaf curling of *Mimosa* plants[4,5]. Similarly, turgor pressure can allow rapid movements by exploiting mechanical instabilities of precisely formed tissues such as in the Venus fly trap and in the explosive dispersal of *Cardamine hirsuta*[6,7]. Alternatively, plant cell walls can passively absorb and release water to cause morphology changes[8,9]. These hygroscopic movements have been demonstrated, for example, in pine cones, wheat awns and ice plant seed capsules[10–12].

Directed hygroscopic movements often arise from the differential expansion of cells within a tissue or parts of cell walls with different material properties. These materials respond to water in different ways to allow, for instance, anisotropic swelling typically resulting in bending or coiling motions[8]. For example, adjacent tissue types with alternating cellulose microfibril orientations generate a bilayer structure to cause bending or twisting motions[13–15]. This can be combined with differential deposition of phenolics. In the curling stems of the resurrection plant, *Selaginella lepidophylla*, different amounts of lignin are deposited on each side of the stem with increased hydrophobicity and elastic modulus observed for tissues where lignin is present. The non-lignified side can therefore absorb more water and deform more easily allowing the plant to unfurl its stems when wet and initiate photosynthesis[16,17]. Similarly, *Erodium gruinum* awns exhibit differential deposition of phenolic compounds in distinct tissue parts affecting the rate of curling along the length of the awn[18].

In addition to material composition, the geometry of cells and tissues can contribute to controlling hygroscopic movements. *S. lepidophylla* cells that swell less tend to have thicker cell walls[16,17]. In the seed capsules of the ice plant, *Delosperma nakurense*, cells with a honeycomb structure expand anisotropically due to their elongated geometry and the arrangement of cell wall layers within them[12]. These examples illustrate that both structural and compositional features are combined to facilitate appropriate and efficient hygroscopic motions, but all rely on heterogeneity of adjacent materials.

The haired fruit of the common dandelion undergoes morphing to open or close its flight-enabling pappus[19–21]. When the hairs are drawn together and the pappus is closed, the fluid dynamics around the pappus are dramatically altered and the dispersal capacity is modified[22]. This allows the plant to tune dispersal by optimising timing and distances in response to environmental conditions. The dandelion pappus changes shape via a hygroscopic actuator at the apical plate of the achene (fruit) that swells on contact with water[19–21].

In addition to hygroscopic absorption of water by cell walls in the apical plate, an alternative pappus morphing mechanism occurs in dandelion pappi relying on the cohesive properties of water droplets. Fine hairs that easily bend are particularly sensitive to the cohesion forces generated by water when it forms a contact point with the solid hairs[23]. Bending of dandelion pappus hairs has been observed before in response to water droplets and may be useful inspiration for engineering precision liquid handling devices[24,25].

While the hygroscopic actuator function of the apical plate has been observed before, its mode of action remains unclear. This actuator is composed of distinct domains originating from the floral podium, vascular bundles and surrounding cortex tissue. We have found that it generates a sophisticated and precisely patterned radial geometry of at least four different tissue types, differential swelling of which enables the reversible angular movement of the pappus hairs. This is more complex than previously described hygroscopic plant actuators which typically rely on one or two tissue layers in planar or cylindrical structures to generate bending or coiling. Radial swelling of the actuator allows the hairs to be pushed both outwards and upwards in contrast to a previous hypothesis suggesting that the hairs are pushed upwards via a lever-like mechanism[20]. Unlike other hygroscopic plant movements, the dandelion makes use of radially symmetric swelling to generate torque, which may help to synchronise movement of the roughly 100 hairs that are organised in a disk like geometry[26].

## Results

**An actuator at the base of the pappus drives morphing**. To investigate the mechanism and dynamics of reversible pappus closure, we imaged dandelion pappi in a bespoke hydration chamber[22] (Fig. 1, Supplementary Movie 1). The extent of pappus closure depends on the amount of water added to the chamber and the pappus reaches a steady state over a period of 30–60 min depending on the dynamics of water addition[22]. In our experiments the pappus angle typically changes by 40–100°.

We examined the apical plate structure at the base of the pappus where the hairs attach that is required for pappus closure and reopening (Fig. 1h). We applied a resin to different parts of the apical plate expecting it to block the ability of the structure to move or swell (Fig. 1a, b, d, e, g, Supplementary Fig. 1). It is possible that water entry into the tissue was also affected. In case (B), the resin was applied to the upper side of the apical plate, in (C) to the lower side and a control set (A) were left unchanged (Supplementary Fig. 1). Blocking the upper side of the apical plate (B) partially prevented pappus closure such that the change in pappus angle was smaller than in free samples but morphing was not completely abolished. However, blocking the lower side (C) almost completely prevented closure (Fig. 1a, b, d, e, g). This indicated that pappus closure does not originate from bending of the hairs themselves, but is largely dependent on the central actuator and particularly on the lower side of it.

An alternative mechanism for pappus closure is via adhesion of the hairs to one another due to surface tension and capillarity. This has been previously demonstrated for dandelion pappi when they hold large droplets of water[24]. To ascertain the role of hair adhesion in the small droplet-derived closing observed here, we removed most of the hairs from dandelion pappi to massively increase the spacing between them (Fig. 1c, f). This would prevent adhesion between neighbouring hairs as demonstrated by Bico et al.[23]. We found that dandelion pappi with just two hairs remaining were still able to close in response to moisture addition (Fig. 1c, f, Supplementary Fig. 2). The dynamics and magnitude were in fact slightly enhanced compared to intact samples (Supplementary Fig. 2). This may be because clusters of hairs normally slightly obstruct one another during motion and

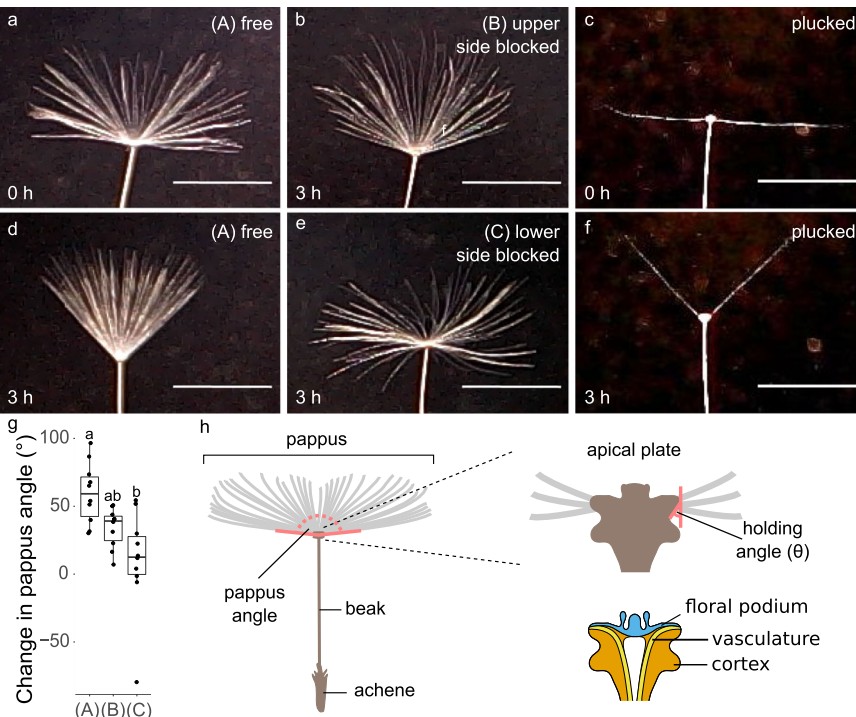

**Fig. 1 The apical plate is required for pappus morphing. a, b, d, e** Apical plate blocking experiments showing **a** a sample before hydration, and **b, d, e** 3 h after wetting. **a** and **d** have no resin applied, **b** has resin applied to the upper side of the apical plate, **e** has resin applied to the lower side of the apical plate. Scale bars are 5 mm. Images in **a** and **d** are of the same sample as each other, images **b** and **e** are different samples from all others shown here. **c, f** Effect of removing hairs on pappus morphing. **c** A sample before hydration with all but two opposite hairs removed, **f** The same sample as in **c** 3 h after hydration. Scale bars are 5 mm and images are representative of n = 10 biologically independent samples for each treatment. **g** The change in pappus angle (between outermost hairs) between dry and wet conditions for apical plate blocking experiment. A, B, and C represent the 'free', 'upper side blocked', and 'lower side blocked' treatments respectively as indicated in panels **a, b, d** and **e**. n = 10 biologically independent samples per treatment, ANOVA $F_{(2,27)}$ = 8.63, p = 0.0013, $\eta^2$ = 0.39. Different letters above two boxplots indicate statistically significant differences between the two corresponding conditions with a two-sided Tukey's HSD at p < 0.001. The centre line is the median, hinges indicate first and third quartiles, and whiskers extend to largest value no further than 1.5 times the interquartile range. **h** Illustrates the morphology of the diaspore, location of the apical plate, and a cross-sectional view of the apical plate indicating some internal structures. Red lines indicate the pappus angle, and holding angle, θ.

removing hairs reduces this effect. These data indicate that surface tension is not involved in this type of pappus closing.

**The pappus actuator inhomogeneously swells to facilitate closure.** As we had confirmed that the apical plate behaved as an actuator, we observed intact apical plates (actuators) and longitudinal half-sections swelling when water was added (Fig. 2a–f, Supplementary Movie 2). The plate is formed of cortex and epidermal cells that crease inwards towards the middle of the structure. The hairs emerge from the epidermal cells on the upper edge (Fig. 2a, b). The cortical tissue is arranged around several vascular bundles and a central cavity (Fig. 2c–f). Situated above all of this is a distinct layer of tissue that originally serves as a nectary and podium for the floral organs before floral abscission and cell death occurs. The dead calyx tissue remains behind and is thereafter named the pappus.

In these circumstances we found rapid radial expansion of the actuator within 2 min of water addition (Fig. 2e–h). Radial expansion was not uniform though as expansion was greatest between the widest points at the lower sides and the narrowest point towards the middle of the structure with an increase of around 40% in distance (Fig. 2g, h). The distance between the outermost points of the floral podium only expanded by around 10% in contrast (Fig. 2g, h). Longitudinal expansion was more homogeneous across the tissue (Fig. 2e, f, i, j).

We observed similar results with environmental scanning electron microscopy in which the microscope chamber pressure

was altered to control the condensation of water droplets on the sample (Fig. 2a–d). In the dry state, most cells appear collapsed and closely packed together but the outlines of some cortical cells towards the lowermost corners of the bulging regions were visible (Fig. 2c, d) and undergo substantial increases in area and circularity (Supplementary Fig. 3). Together these results indicated that the apical plate expands when wet in a heterogeneous manner.

**The actuator has distinct domains with differential cell wall composition.** As the floral podium of the apical plate expanded in a different way to the lower cortical regions, we hypothesised that these tissues form a classic hygroscopic bilayer where the floral podium tissue is less able to swell than the lower cortex. This would require different material composition or properties. We found that the floral podium did not appear to be lignified in contrast to our initial expectations (Fig. 3a). That layer did autofluoresce when illuminated with UV light, however, indicating phenolic compounds were likely to be present (Fig. 3b). This autofluorescence increased at higher pH suggesting the presence of ferulic acid (Fig. 3c, d).

We found other regions of the apical plate with distinct cell wall compositions (Fig. 3). Phloroglucinol-HCl stained most cell types but was particularly enriched in the vascular bundles indicating the presence of lignin as is common for xylem and associated fibres (Fig. 3a). An intriguing lipid-rich region was also

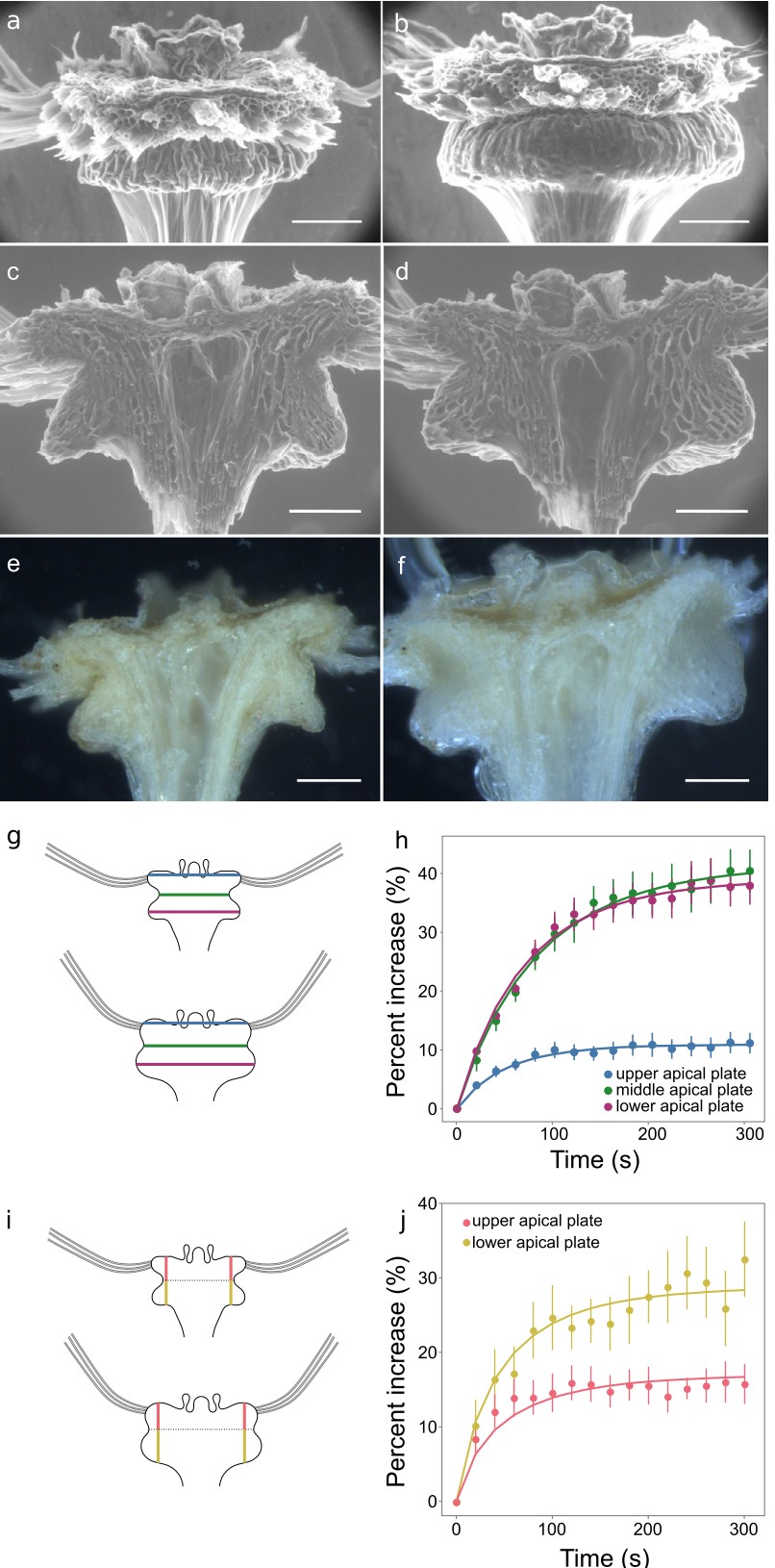

**Fig. 2 Apical plate expansion during hydration. a–d** Environmental SEM images of apical plate samples before, **a**, **c**, or after, **b**, **d**, hydration. **a**, **b** Intact sample with the majority of hairs removed. Images are representative of $n = 3$ samples. **c**, **d** Longitudinal half-sections. Images are representative of $n = 10$ samples. **e**, **f** Light microscopy images of longitudinal half-sections of an apical plate imaged before **e**, and after **f**, hydration. Images are representative of $n = 19$ samples. **g** Schematic illustrating the locations of radial measurement, **h** Quantification of radial expansion in half-sections imaged with light microscopy, $n = 10$, error bars are standard error of the mean. **i** Schematic illustrating the locations of longitudinal measurement, **j** Quantification of longitudinal expansion in half-sections imaged with light microscopy, $n = 10$, error bars are standard error of the mean. Scale bars are 100 μm.

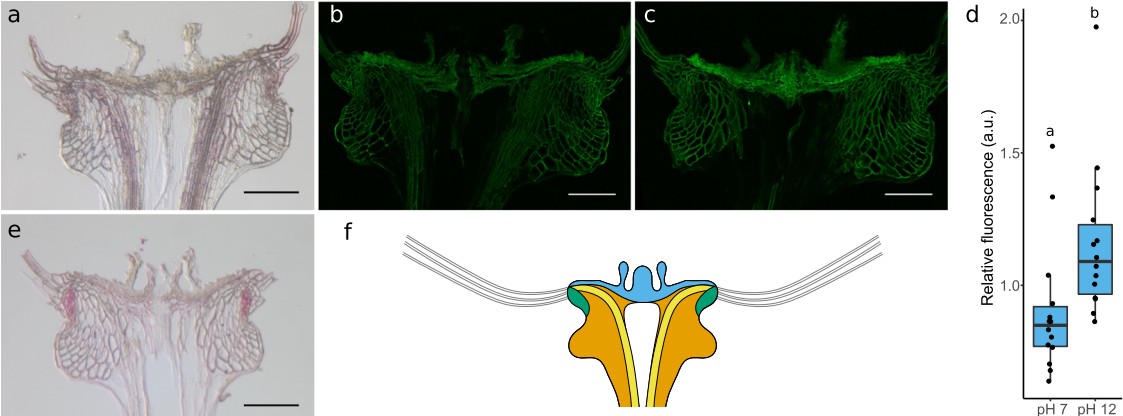

**Fig. 3 Apical plate composition. a** Phloroglucinol-HCl staining for lignin, image is representative of $n = 9$ samples, scale bar is 100 μm. **b, c** UV autofluorescence at **b** ph 7, and **c** pH 12. Images are representative of $n = 14$ samples, scale bars are 100 μm. **d** Quantification of autofluorescence signal, $n = 14$ per treatment. Different letters above boxplots indicate significant differences with a two-sided Wilcoxon signed rank test, $p = 0.0001$. The centre line is the median, hinges indicate first and third quartiles, and whiskers extend to largest value no further than 1.5 times the interquartile range. **e** Sudan Red 7b staining for lipids, images are representative of $n = 10$ samples, scale bar is 100 μm. **f** Illustration of the different regions of the apical plate with floral podium in blue, cortex in orange, vasculature in yellow and side regions in green.

revealed around the upper sides of the cortex adjacent to the attachment site for the hairs by staining with Sudan Red 7b (Fig. 3e). Using Raman spectroscopy we found distinct intensity peaks around $1600 \, \text{cm}^{-1}$ indicating the presence of phenolic compounds[27] in all regions, and, in particular, in the side regions (Supplementary Fig. 4). A unique peak at around $1630 \, \text{cm}^{-1}$ in the floral podium may correspond to ferulic acid[28]. These data indicate that there are at least four domains of the actuator with distinct cell wall compositions or arrangements: the floral podium, the vasculature, the lipid-rich sides, and the remaining cortex cells (Fig. 3f).

**Outward radial expansion drives pappus closure**. As the apical plate structure was more complex than we initially imagined, we investigated the tissue expansion in the four distinct domains (Fig. 4). The autofluorescence of the phenolics was captured in longitudinal half-sections by taking high resolution z-stacks using laser confocal scanning microscopy (Fig. 4a, b). The same samples were imaged when completely dry and when tissues had reached a steady state after being saturated with water. Salient landmarks, such as cell corners and small protrusions, that could be clearly identified in both dry and wet images were annotated.

We selected a central landmark close to the middle point of the floral podium as a reference, reflecting the fact that the structure is approximately radially symmetric. Using this, the relative displacement of landmarks was calculated (Fig. 4e). The displacement map highlighted that the lower cortical regions displaced in a lateral direction from the centre and that points around the sides near where the hairs attach generally also moved outwards but also curved upwards. The vascular bundles showed limited displacement from the centre and where displacement did occur it was also largely in a radial or downward direction (Fig. 4e). This tissue expansion pattern contradicts the previous hypothesis that the lower bulges of the cortex push upwards on the base of the hairs to lever them upright[20]. Instead, the radial swelling of the tissue unfolds the crease in the middle of the structure to draw the hair attachment sites outwards and rotate them into an upright position.

As point displacement reflects cumulative changes across the whole tissue, local expansion rates were also calculated to help understand the mechanism behind the anisotropic swelling (Fig. 4d, e). Triangles were formed between neighbouring landmark points to allow

calculation of relative expansion. These triangles were assigned an identity based on the tissue region they mostly overlapped with (Fig. 4c).

Our previous measurements of radial expansion indicated minimal expansion for the floral podium tissue compared to the lower regions while longitudinal expansion was more uniform in the upper and lower halves (Fig. 2e–j). The more detailed characterization of regional expansion demonstrated that all tissues expand to a roughly similar degree (50–90%), except for the vasculature, which shows reduced expansion (30%) (Fig. 4f). This suggests that the vascular bundles may act primarily as a resisting tissue rather than the upper floral podium layer. As the vasculature is embedded within the actuator, the reduced capacity for expansion would anchor the adjoining tissues and cause anisotropic expansion overall.

**A mechanical model of pappus morphing**. We hypothesized that pappus closure arises from differential water absorption or swelling properties of the different tissue regions. We also expected that the geometry and arrangement of these tissues would impact on the changing angle of the hairs. To understand how these components work together to allow pappus closure, we constructed a simplified computational mechanical model of the pappus actuator, employing the Finite Element Method. We considered the actuator as a two-dimensional, isotropic, linearly elastic system that undergoes shrinkage due to loss of water.

In the model, the apical plate is divided into four regions: floral podium, vasculature, sides, and cortex (Fig. 5a). The tissue types were arranged according to measurements of the real dandelion pappus actuator in the hydrated configuration (Table 1). The model was allowed to shrink as if water was evaporating from the structure to generate the dry state version (Fig. 5e). To account for the differences between a 2D model of a 3D phenomenon and for the simplified tissue geometry, a boundary condition was imposed that the base of the vascular bundles displaced by a fixed amount (Table 1). The angle, θ, in the dry state represents the pappus holding angle (Fig. 5e) and was the main prediction of the model. This was defined as the angle between the vertical, and the line between the upper corner of the apical plate and the lowest point on the lateral edge of the side region. We compared θ to the equivalent measurements on the imaged longitudinal half-sections.

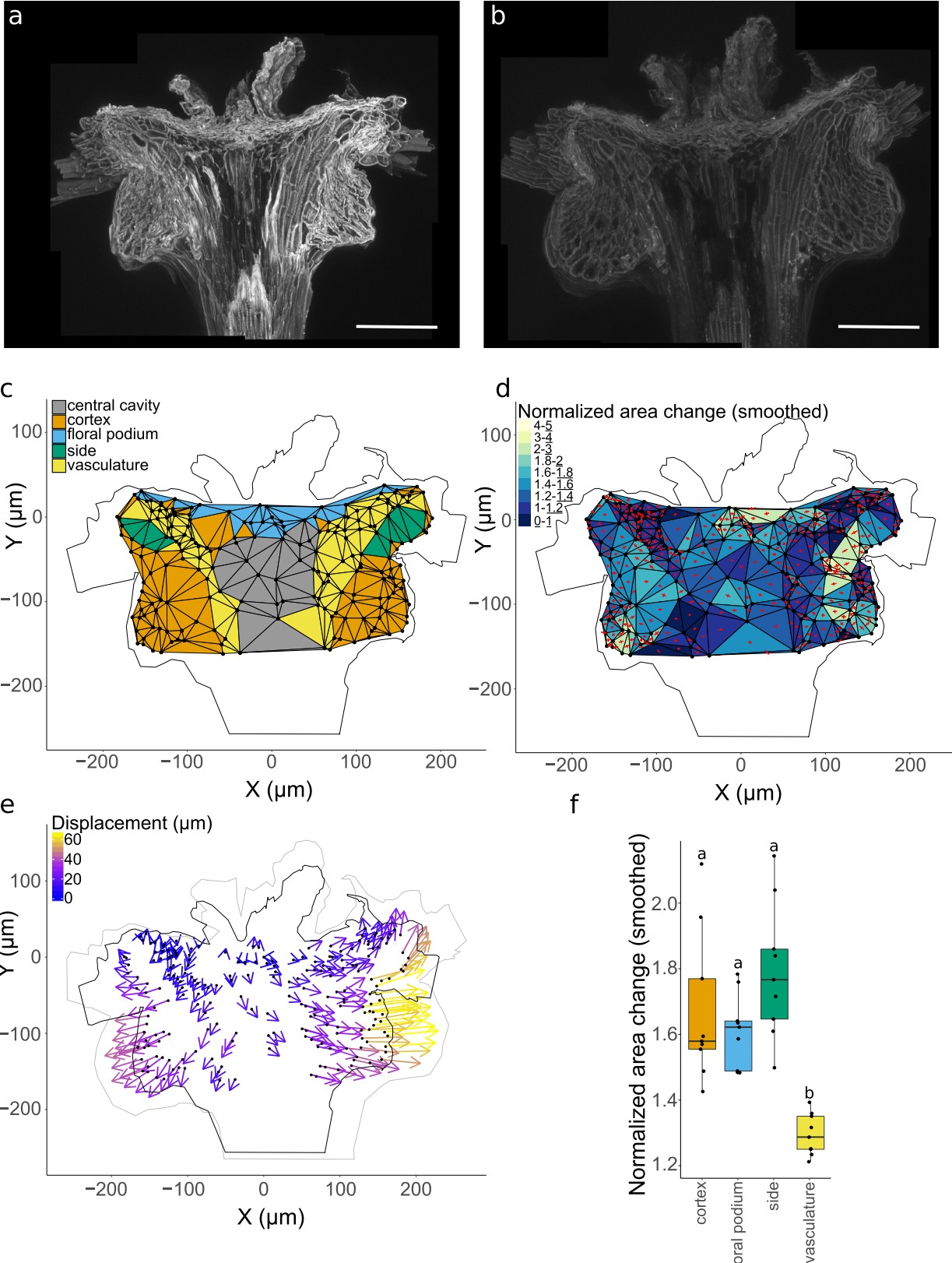

**Fig. 4 Apical plate regional expansion and displacement. a**, **b** Confocal images of apical plate autofluorescence in dry, **a**, and hydrated, **b**, states, scale bars are 100 μm, and images are representative of $n = 9$ biologically independent samples. **c** Landmarks and triangulation with regions annotated. **d** Smoothed change in area of each triangle (wet area/dry area). Red crosses indicate principal directions of expansion. **e** Displacement of landmarks relative to upper central part of the apical plate. **f** Change in area by region. ANOVA $F_{(3,32)} = 14.44$, $p < 0.0001$, $\eta^2 = 0.58$. Different letters above two boxplots indicate statistically significant differences between the two corresponding conditions with two-sided Tukey's HSD at $p < 0.01$. The centre line is the median, hinges indicate first and third quartiles, and whiskers extend to largest value no further than 1.5 times the interquartile range. $n = 9$ biologically independent samples for all panels.

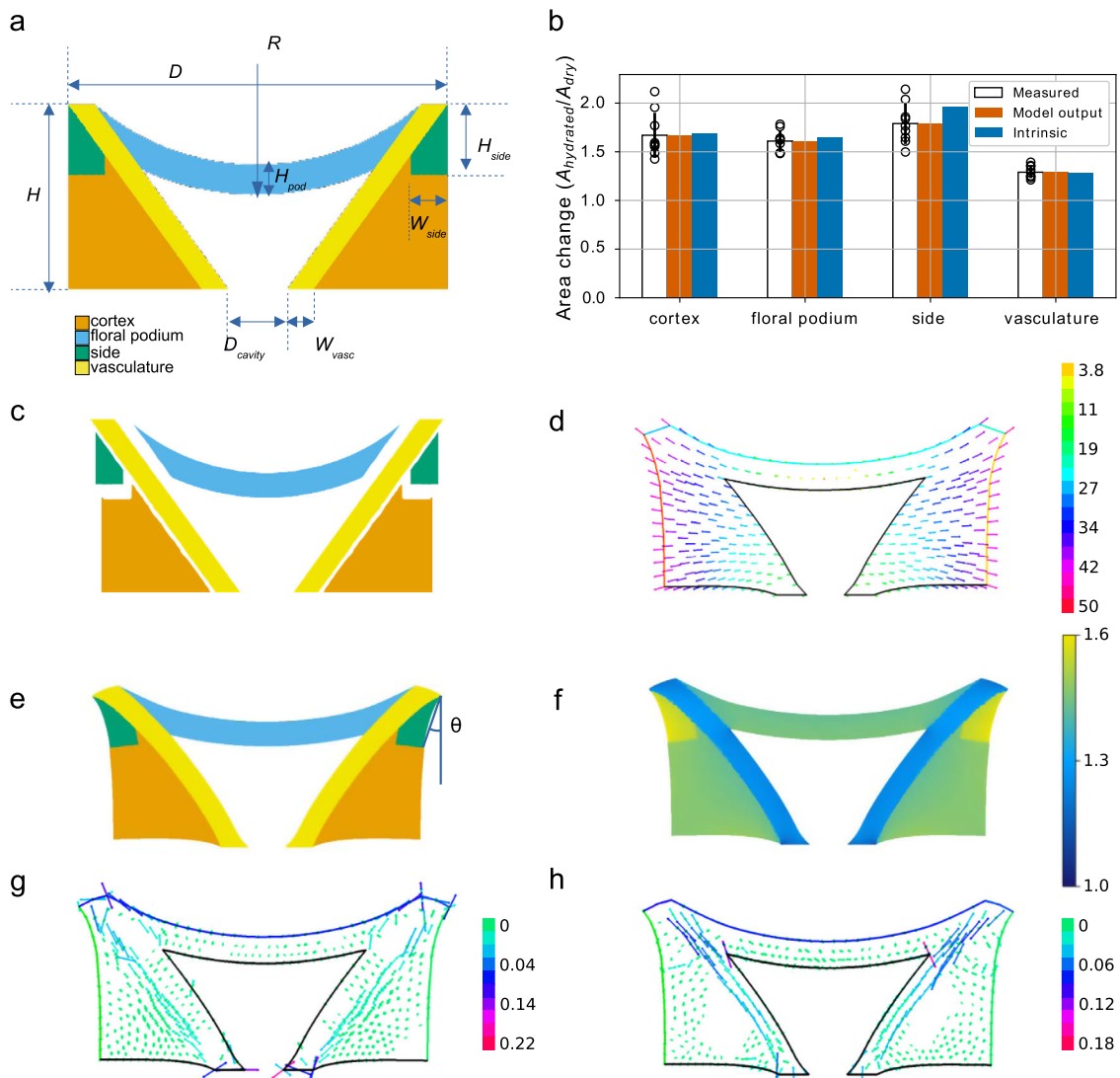

**Fig. 5 Computational modelling of apical plate behaviour. a** Geometry of the hydrated apical plate with geometrical parameters annotated. The four regions are: floral podium (blue), vasculature (yellow), sides (green), cortex (orange). **b** Observed (as seen in panel **e**) and intrinsic (as seen in panel **c**) area changes by region compared to the measured area changes. Black bars indicate standard deviation. **c** Dry state of each region arising from different intrinsic swelling property in a hypothetical setting in which the regions are not attached to each other. **d** The displacement field relative to the centre of the floral podium. **e** The dehydrated apical plate with regions annotated. Differential intrinsic swelling causes changes in shape due to the fact that they are adhered together. Dehydration generates the holding angle, $\theta$, which is annotated. **f** Local changes in area across the apical plate. $n = 9$ biologically independent samples for measured data and is the same dataset as Fig. 4f. **g** Principal tensile stress in the dehydrated state. **h** Principal compressive stress in the dehydrated state.

| Table 1 List of measured geometrical parameters used to define the model. | | | | |
| --- | --- | --- | --- | --- |
| Parameter | Symbol | Value | Units | Uncertainty (%) |
| Diameter of apical plate | $D$ | 491.0 | μm | 0.07 |
| Height of apical plate | $H$ | 240.0 | μm | 0.15 |
| Radius of curvature of floral podium | $R$ | 363.0 | μm | 0.14 |
| Height of floral podium | $H_{pod}$ | 38.8 | μm | 0.25 |
| Height of side regions | $H_{side}$ | 91.4 | μm | 0.14 |
| Width of side regions | $W_{side}$ | 47.6 | μm | 0.14 |
| Width of vasculature | $W_{vasc}$ | 34.3 | μm | 0.16 |
| Diameter of cavity at base | $D_{cavity}$ | 74.8 | μm | 0.20 |
| Vascular displacement from wet to dry configuration | $d_{vasc}$ | 15.3 | μm | 0.22 |

To focus on the main mechanical design features of actuation, we assigned homogeneous material properties to each region taking epidermis and cortex as one tissue type (Table 2). Each region was assigned elastic properties and an intrinsic swelling property that quantifies relative changes in area upon dehydration if the region were isolated from neighbouring regions (Table 2, Fig. 5c). We measured or chose plausible values for all parameters other than the four intrinsic swelling parameters (Tables 1 and 2). All geometric parameters were measured from images of apical plate sections. The cells of the apical plate appeared similar to wood cells with thickened secondary cell walls and lignification. As a result, we used a Poisson ratio typical for woody tissues[29] (Table 2). The elastic properties were prescribed according to the measured density of cell wall material of each part (Table 2) based on measurements with atomic force microscopy indicating that the wall-scale Young's moduli were similar across regions (Supplementary Fig. 8). Mechanical

**Table 2 List of measured, estimated, and fitted parameters used to define material properties in the model.**

| Parameter | Symbol | Origin | Value | Units | Uncertainty (%) |
|---|---|---|---|---|---|
| Density of cortex tissue relative to density of vasculature | $\rho_{cort}/\rho_{vasc}$ | Measured | 0.76 | none | 0.14 |
| Density of floral podium tissue relative to density of vasculature | $\rho_{pod}/\rho_{vasc}$ | Measured | 1.16 | none | 0.09 |
| Density of side tissue relative to density of vasculature | $\rho_{side}/\rho_{vasc}$ | Measured | 1.04 | none | 0.16 |
| Poisson ratio | $\nu$ | Estimated | 0.29 | none | NA |
| Swelling factor of cortex | $s_{cort}$ | Fitted | 0.46 | none | NA |
| Swelling factor of floral podium | $s_{pod}$ | Fitted | 0.44 | none | NA |
| Swelling factor of side | $s_{side}$ | Fitted | 0.57 | none | NA |
| Swelling factor of vasculature | $s_{vasc}$ | Fitted | 0.24 | none | NA |

conflicts may arise between regions if they have different intrinsic swelling properties, leading to observed swelling that differs from intrinsic swelling (Fig. 5b, c, e). The unknown intrinsic swelling parameters were fit to observations by optimising expansion in the model to the measured regional expansion from our landmark triangulation analysis (Fig. 5b). We call the model together with this set of parameters the 'reference model' (Fig. 5).

In Fig. 5b we represent the measured area changes for each region, the optimised simulated area changes as well as the corresponding intrinsic swelling. Notice the difference between the model output area changes and the intrinsic swelling per region. The regions would not change shape if they would morph independently with their own intrinsic swelling as can be seen in Fig. 5c. However, due to the fact that regions are attached to one another, mechanical conflicts arise from the differential swelling between regions and gives rise to the change in shape of the structure (Fig. 5e), such that the hypothetical intrinsic and actual observed area changes are different. These mechanical conflicts are visible when plotting mechanical stress patterns in the dry state (Fig. 5g, h) and appear localised to vasculature and a neighbouring band along cortex and sides; vasculature is longitudinally compressed by relatively higher shrinkage in the neighbouring band, while, conversely, cortex and sides are under tensile stress parallel to the axis of vasculature due to reduced shrinkage in the vasculature. Higher tensile stress in the dehydrated versus hydrated state of the cortex is supported by gaping of incisions to the apical plate in dehydrated samples (Supplementary Fig. 9), and contradicts a previous model of this phenomenon that implicitly assumes higher stress in hydrated samples[20].

The holding angle, $\theta$, obtained as an output of the reference model is about 20° (compared to a measured $\theta = 36° \pm 6.7$). The displacement field relative to the centre of the podium which relates the dry to the wet state of the reference model (Fig. 5e) shows similar radial displacements as was measured and shown on Fig. 4e. Additionally, the local area change and displacement maps that we obtain from the model (Fig. 5b, d, f) are comparable both quantitatively and qualitatively to the measured versions (Fig. 4d, e). Therefore, the reference model sufficiently recapitulates the observed behaviour of the pappus actuator.

**The intrinsic swelling properties and dimensions of the apical plate are important for actuator function**. To understand the reference model further, we carried out a one-factor-at-a-time sensitivity analysis to see which features of the model are most important for actuator behaviour (Fig. 6a, Supplementary Fig. 5). We took the reference model and varied one parameter around its reference value while keeping all others unchanged. We monitored the relative change in the holding angle, $\theta$, with respect to the relative change of the parameter in question (Supplementary Fig. 5). The sensitivity of $\theta$ to a given parameter is the ratio of these relative changes (see Supplementary Note 1), and reflects

the direct effect of this parameter on $\theta$ (Fig. 6a). The fact that sensitivities defined this way are non-dimensional, normalised quantities, allows us to compare sensitivities among each other. In particular, examining the effect of each model parameter on the output $\theta$, we found that the fitted intrinsic swelling capacity of each tissue had the greatest effect on holding angle (Fig. 6a, Supplementary Fig. 5). Increasing the swelling capacity of the cortex or side regions greatly increased $\theta$ while the opposite was true for the vasculature and floral podium, which showed increased $\theta$ values when swelling capacity was decreased. For geometrical changes, the sensitivity analysis highlighted the overall dimensions in the horizontal and vertical directions ($D$ and $H$) and the radius of curvature of the floral podium ($R$) as having substantial effects on the holding angle change (Fig. 6a, Supplementary Fig. 5).

In theory, our model predicts how angle change varies with each of the actuator geometrical parameters. However, these geometrical parameters are not independent and co-vary in biological specimens (Supplementary Fig. 6). To test the validity of our model, we used the sensitivity analysis to compute the correlation between $\theta$ and geometrical parameters given the measured natural correlations between geometrical features observed in our specimens (Fig. 6b, Supplementary Fig. 6). A full description of the method can be found in Supplementary Note 1. The predicted correlation between $\theta$ and geometrical parameters was compared with measured correlations between these features. The confidence intervals were generally large for most measured correlations, but for those that were statistically significant, the predicted correlation coefficients fell within the expected range for all but one ($H_{side}$) parameter (Fig. 6b). We surmised that the cause of this large negative correlation between $H_{side}$ and $\theta$ is due to the way in which $\theta$ is measured. Note that $\theta$ was determined as the angle between the vertical, and the line formed between the upper corner of the apical plate to lateral edge at the lowest point of the side region, which has height $H_{side}$ in the dehydrated state (Fig. 5e). Therefore, when the parameter $H_{side}$ was varied in the sensitivity analysis, the line giving the definition of the holding angle $\theta$ also changed. We therefore believe that the large negative correlation between $H_{side}$ and $\theta$ in the sensitivity analysis is largely due to the changing location of this measurement line.

As in the one-factor-at-a-time sensitivity analysis, the sensitivity analysis accounting for natural co-variations highlighted the importance of $H$ and $D$ in positively regulating $\theta$ (Fig. 6b). $H_{pod}$ was less significant in this context, while $H_{side}$ appeared much more important for determining $\theta$. In conclusion, this analysis predicted correlations between $\theta$ and geometrical parameters, in fair agreement with experimental measurements (Fig. 6b).

**The material properties of each domain affect actuation**. To understand further the extent to which material properties affected actuator behaviour, we created hypothetical scenarios in which certain regions were made of alternative tissue types

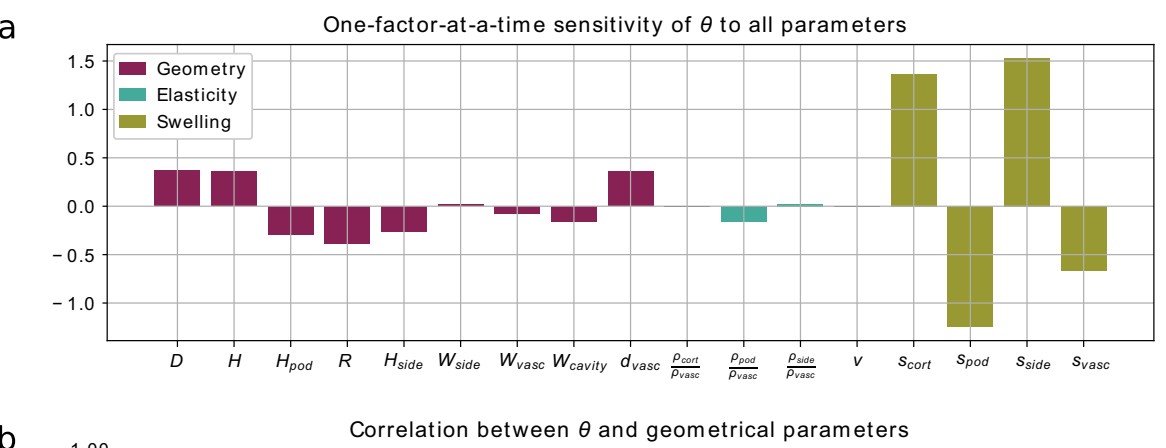

a — One-factor-at-a-time sensitivity of $\theta$ to all parameters

b — Correlation between $\theta$ and geometrical parameters

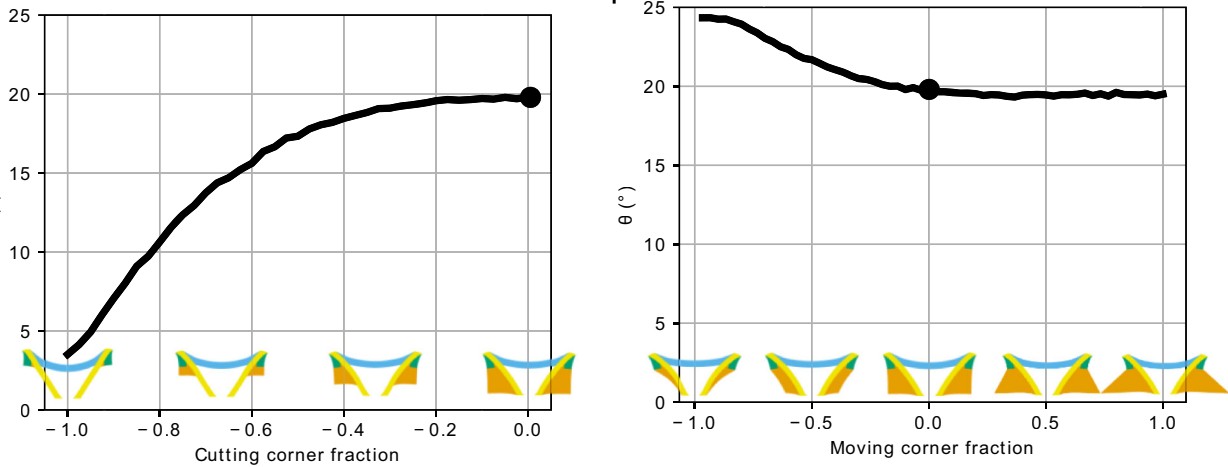

**Fig. 6 Predictions of the output holding angle ($\theta$) resulting from different perturbations of the model. a, b** Perturbations of the parameters. **a** The partial sensitivity of $\theta$ to input parameters obtained by an OAT (one at a time) sensitivity analysis. **b** The computed (predicted) correlation of geometrical input parameters to $\theta$, when correlation between input parameters is taken into account, are compared to the measured correlation coefficients. Black bars denote 95% confidence intervals around the measured value. Measured correlation coeffiecents are based on $n = 9$ biologically independent samples and are a subset of the same data represented in Supplementary Fig. 6a. **c–f** Perturbations of the geometry of the reference model (marked by a black circle) by adding (positive values) or removing (negative values) cortex material. **c** The central cavity is filled horizontally. **d** The central cavity is filled vertically. **e** Cortex material is removed from the corner of the actuator vertically. **f** Cortex material is removed from or added to the actuator by moving the corners horizontally.

(Supplementary Fig. 7). We found that substituting the material of each region with cortex-like material generally reduced the magnitude of the holding angle, $\theta$ (Supplementary Fig. 7a). Despite this, we observed that the model was still quite robust and achieved moderate angle changes (>14°) when all regions contained cortex-like tissue, provided the vasculature retained its original properties of higher stiffness and minimal swelling capacity (Supplementary Fig. 7a). When the side regions were the only contrasting tissue to cortex, only a small $\theta$ was observed but when combined with vasculature in its natural state, an enhanced $\theta$ was observed comparable to the reference model (Supplementary Fig. 7a). This indicates that the distinctive vasculature material properties (low swelling capacity and moderately high stiffness) relative to the other regions are most important for actuator behaviour. In this case, substituting the floral podium for cortex tissue had no substantial effect on the resulting holding angle (Supplementary Fig. 7a) probably because the material properties (in particular the swelling factor) of these two regions are quite similar (Table 2).

We then substituted each region with material with side region-like properties (Supplementary Fig. 7b). Making the vasculature even more swellable than the cortex (swelling factor 0.57 vs 0.46, Table 2) entirely abolished the actuator function and if, in addition, the floral podium was also substituted with side material, the actuator began to invert and generate negative $\theta$ values (−8.4°) (Supplementary Fig. 7b). Substituting only the floral podium for the side material that swells more also reduced the actuator function, though to a lesser extent (12.8°). This indicates that though primarily the vasculature properties must contrast with the cortex, the floral podium also plays a resisting role. Interestingly, the actuator function could be enhanced by replacing cortex with side material indicating that the actuator may not be perfectly optimised for maximising holding angle changes (Supplementary Fig. 7b).

We also attempted to substitute regions with the vasculature tissue type to assess the effect of reducing swelling capacity and, for some regions, increasing stiffness (Supplementary Fig. 7c, Table 2). Applying this tissue type to either cortex or side regions caused only a modest reduction in $\theta$ indicating that a structure possessing either contrasting tissue type adjacent to the vasculature can be almost sufficient (Supplementary Fig. 7c). However, this relied on also having a floral podium that could not swell easily (i.e., replaced with vasculature material). Without this restraint, the actuator almost completely failed. If the floral podium was replaced with vasculature but the other regions remained normal, the largest holding angle changes were observed compared to any other scenario including the reference model (Supplementary Fig. 7c).

Together, these data suggest that for the apical plate to function best as a hygroscopic actuator, the vasculature and floral podium must be unable to swell easily while the cortex must easily swell. Highly swelling side regions can enhance the behaviour of the actuator (though increasing the size of that domain can be detrimental) but are not essential to broadly capture the actuating function (Supplementary Fig. 7).

**The arrangement of tissues around the central cavity is essential for changing holding angle.** To explore the importance of the observed apical plate geometry, we made some modifications to the geometry of the computational model and assessed the predicted holding angle, $\theta$ (Fig. 6c–f). We found that the central cavity between the vascular bundles and beneath the floral podium was necessary to allow substantial angle changes (Fig. 6c, d). Progressive filling of the cavity with cortical tissue either laterally or longitudinally correspondingly reduced $\theta$ (Fig. 6c, d)

We also test the role of the cortex tissue surrounding the vasculature (Fig. 6e, f). We found that altering the size of the lower bulges in the lateral direction had very little effect on $\theta$ (up to 5° difference from the reference model) and reducing the tissue to only a thin layer around the vasculature was actually able to slightly increase $\theta$ (Fig. 6f). In contrast removing the cortex tissue longitudinally caused a dramatic reduction in the angle change (a decrease of 14° from the reference model) (Fig. 6e). This indicates that the quantity of cortex tissue present around the vasculature is not important, but that the connection between the side regions and the whole length of the vasculature must be maintained (Fig. 6e). Together, these results indicate that the geometry we observe is almost optimal for pappus closure in comparison to the majority of modifications we made. They also highlight a conical vasculature-cortex bilayer structure at the core of the actuator's functioning: both removing the outer cortex layer adjacent to the vasculature (Fig. 6e) and adding a third internal cortex layer in the central cavity (Fig. 6c, d) considerably impairs the morphing capability of the structure.

## Discussion

Our data indicate that the balance of resisting and swelling tissues is carefully arranged in the dandelion actuator to allow functional angle changes (Figs. 3, 6 and Supplementary Fig. 7). The vasculature and floral podium together provide a resisting framework anchoring the surrounding swellable tissues that provide the majority of the swelling motion. Additionally, the central cavity is essential to provide space for the other regions to contract into when drying (Fig. 6c, d).

This arrangement allows precise radial swelling that is not seen in other hygroscopic movements. While there are other systems that involve the bending of filamentous structures that are radially arranged (such as in the unfurling of *S. lepidophylla* stems[17] and in the bending pedicels of carrot umbels[15]), these rely on separate bending of each stem individually. For the smaller dandelion pappus with exceptionally fine hairs[26], it is possible that a single actuating structure is more efficient and allows better co-ordination of motion. This might prevent tangling or breakage of hairs if they were misaligned during bending. We expect that this is not unique to the dandelion as many other Asteraceae exhibit hygroscopic opening and closing of their pappi[19,21]. Additionally, many plant structures are inherently radial so it is likely that a pre-existing structural arrangement of vascular fibres and other tissues could be easily selected for during evolution to facilitate biomechanical movements.

The hygroscopic plant movements that have so far been described involve a combination of structural and compositional features of cell walls that result in either bending, twisting or coiling motions. These typically rely on a bilayer structure consisting of a swelling tissue or cell wall compartment and an adjacent resisting element. The dandelion apical plate can be considered as a variation of bilayer structure since swelling regions must be connected to regions that either swell less or swell in a different direction. The radial motion, however, means that a more complex structure is required consisting of at least three tissue types (Fig. 3). In our model, structures with only two tissue types performed less well than the reference model (Supplementary Fig. 7). Even in cases where two tissue types gave a $\theta$ value comparable to the reference model (Supplementary Fig. 7b - sssv, 7c – vvsv), there were circumstances where adding a third tissue type could further enhance $\theta$ beyond the level of the reference model (Supplementary Fig. 7b - spsv, 7c - cvsv).

Assigning reduced swelling properties to the floral podium enhanced the pappus holding angle changes in the model (Fig. 6, Supplementary Fig. 7) (though note that increasing the height or density of the podium reduces $\theta$, Fig. 6a). This indicates that having a three-part structure in which a bilayer is combined with another resisting material on an intersecting plane further constrains some of the swelling to enhance anisotropic motion. This is more reminiscent of the cup-shaped thickening in the individual cells of fern sporangia that collectively resist bending during dehiscence and allow directed opening of the spore capsule[30]. It is possible that in the 2D model, the floral podium acts primarily to connect the tissues and keep the two sides separate. A 3D computational model may change this requirement as the remaining tissue would be radially connected.

Our data and modelling give some hint as to the mechanism behind the differential swelling properties of each region. The strong lignification of the vascular tissue (Fig. 3a) suggests that these cells are highly hydrophobic, which corresponds to the minimal expansion observed. Ferulates in the floral podium (Fig. 3b–d) also indicate a hydrophobic material though it is also likely that high density of cell walls (Table 2) causes that layer to act as a partially resistive tissue. Though the cortical cells do appear to be somewhat lignified (Fig. 3a, b), the cells are larger with significant spaces between them (Fig. 2c–f, Table 2). This may mean that for a given proportional swelling of those cell walls there are greater absolute levels of expansion at the tissue level compared to smaller, denser cells in other tissues. These cells increase substantially in area, and also unfold and become more rounded as they expand (Supplementary Fig. 3), whereas the smaller, narrower vascular cells remain almost unchanged in shape and size when they are wet (Fig. 4).

The side regions contain a lipid-rich material (Fig. 3e), which might be suberin. While suberin is classically associated with the hydrophobic Casparian strip of the root endodermis and abscission zones[31], the side cells of the dandelion apical plate expand considerably with wetting. While this appears contradictory, the relative hydrophobicities of cell wall materials are not well understood and in fact Casparian strip water impermeability appears to rely more on ferulate components than on waxes[32].

A feature of other hygroscopic systems is that they frequently rely on inherent anisotropic swelling of the cell walls by orientating cellulose microfibrils[8,9]. In the dandelion apical plate, we cannot rule out intrinsic swelling anisotropy as a possibility. However, we find from our computational model that intrinsic anisotropy is not necessary for a functional model. Simply juxtaposing tissue types with differential isotropic swelling capacities and stiffnesses that arise solely from cell density recapitulates the behaviour of the actuator, notably the observed area changes (Fig. 5). While there are some small differences between the model and experiments, this might be due to 2D modelling of a 3D phenomenon and even the 20° change in angle observed in the model would substantially impact flight behaviour[22].

The hygroscopic actuator underlying dandelion pappus morphing is made of non-living cells, whose cell walls have differential water-dependent expansion and drive the movement of the pappus hairs. It is a previously uncharacterised type of biological hinge, which is a radial, tubular actuator that can coordinate collective movement of hairs positioned on a planar disk at the periphery, like an umbrella. The dandelion actuator has design elements resembling bilayer structures, yet it is distinct from planar or cylindrical bilayer actuators that bend or fold, which have been well explored in biomimetic soft robotics. From intestines and bladder to sea anemones, tubular actuators are abundant in living systems and represent a next frontier for biomimetic soft robotics. The simple design requirement of the dandelion actuator makes it likely possible to be fabricated at micro-scale or smaller, and may be applied to the creation of novel animated functional materials.

## Methods

**Plant material.** *Taraxacum officinale* agg. samples were collected in Edinburgh (55.922502, −3.170236) and grown as described previously[26]. Seeds were germinated in Petri dishes containing distilled water for two weeks in 16 h light 25 °C/8 h dark 23 °C conditions (light levels were 100 µmol m$^{-2}$ s$^{-1}$). Seedligs were transplanted to $7 \times 7 \times 8$ cm$^3$ pots containing 60% v/v Levington's F2 + S (Everris), 24% v/v standard perlite (Sinclair), 16% v/v sand with 0.3 g L$^{-1}$ Exemptor (Everris). Plants were grown in 16 h light / 8 h dark conditions in a controlled environment room (100 µmol m$^{-2}$ s$^{-1}$ 21 °C) for four weeks and then transplanted into 4 L pots containing 83% v/v medium peat (Clover), 21% v/v sand, 3 g L$^{-1}$ garden limestone (Arthur Bowers), 1 g L$^{-1}$ Osmocote Exact Standard 5-6 months (Everris), 0.4 g L$^{-1}$ Exemptor (Everris). These 4 L pots were grown in a glasshouse with ambient light supplemented to ensure a 16 h day with minimum light intensity of 250 µmol m$^{-2}$ s$^{-1}$ and minimum temperatures of 21 °C during the day and 18 °C during the night. All samples originated from the same original individual and were grown in the greenhouse for two generations. The offspring of these were used for experiments. Diaspores used were assumed to be genetically identical as this subspecies reproduces apomictically. For all experiments, each sample was a from a different fruit selected from different plants. The same subspecies (a member of section 'Taraxacum'), was used for all experiments.

**Moisture chamber imaging.** A bespoke moisture chamber was set up to release small mist droplets into a air-tight box containing dandelion fruits. This was assembled as described in ref. [22]. A 70 L airtight box was used and a small hole made to pass cables through, which was then sealed with silicone sealant. An ultrasonic humidifier was used for water release and small USB microscopes (Maozua, USB001) for imaging. A datalogger (Lascar Electronics, EL-GFX-2, 0.1 °C resolution for temperature, 0.1% resolution for relative humidity, measurements taken every 10 s) was included in all experiments to monitor temperature and relative humidity. Samples were fixed in place by embedding in plasticine or by crafting small holders out of aluminium foil wrapped around the achene. For all experiments, except Supplementary Video 1, moisture was added to the chamber for 20 min and pappi allowed to absorb water and equilibrate with their surroundings within the sealed chamber. Images were captured at the start of the experiment before water was added and after three hours. For Supplementary Video 1, the humidifier was left on for the duration of the imaging experiment and images captured every 30 s.

For the apical plate blocking experiments, methacrylate nail polish was carefully applied using a sewing pin and subsequently cured by exposing to UV light. For blocking of the lower side of the apical plate the hairs of dandelion pappi were temporarily tied loosely together using cotton thread to allow access. Control pappi had hairs temporarily tied together but no nail polish applied.

To test the effect of increasing hair spacing, pappi were either left intact as a control treatment or had the majority of hairs carefully removed using fine forceps. Two hairs were allowed to remain that were approximately opposite one another.

**Microscopy and histology.** For longitudinal half-sections, dandelion pappi with hairs mostly removed were directly embedded into paraffin wax without any fixing or infiltration. A microtome was used to slice away material and the cut surface examined periodically under a microscope. Once the central cavity and the bulge at the centre of the floral podium (the stylopodium) were visible, the apical plate was considered to be cut in half. The majority of the wax was cut away manually and the remaining half section was briefly submerged in Histoclear to dissolve any remaining wax.

To visualise cell wall materials using cell wall stains and autofluorescence, dandelion pappi were progressively infiltrated with water followed by ethanol, Histoclear and paraffin wax. Longitudinal sections of 10 μm thickness were made using a microtome and sections were dewaxed using the inverse infiltration procedure. For lignin staining, 2 parts of 3% (w/v) phloroglucinol dissolved in absolute ethanol and combined with 1 part concentrated HCl according to ref. [33]. Sections were imaged immediately using a stereomicroscope. Lipid staining was carried using 0.1% (w/v) Sudan Red 7B dissolved in PEG-300 and combined 1:1 with 90% glycerol according to ref. [34]. Staining was carried out for 16 h before washing with water, mounting in glycerol, and imaging using a stereomicroscope. Autofluorescence images were acquired using a confocal microscope with an excitation wavelength of 405 nm and emission acquisition from 480–600 nm. For ferulic acid identification, paired sections were used in which two adjacent longitudinal slices close to the medial section of each apical plate sample. These were imaged using a 10× objective at either pH 7 (distilled water) or pH12 (0.01 M KOH) for 15 min[35].

To image apical plate expansion, apical plate half sections were lightly glued to a glass slide with a small dot of epoxy resin and a coverslip overlaid. Images were taken of dry samples and then after wetting by pipetting distilled water underneath. For time course imaging, a stereomicroscope was used and images captured every 20 s. For landmark annotation, a 63× objective was used to image autofluorescence of the cut face of the samples. Tiled z-stacks were acquired and maximum projections were later stitched together of each sample when dry and wet after allowing the tissue to reach a fully expanded hydrated state (around 30 min).

Apical plate expansion was also imaged in an environmental scanning electron microscope (ESEM) (FEI Philips XL30). Half-section samples were prepared as described above and placed onto an adhesive carbon pad to mount on the ESEM stage. The Peltier stage was maintained at 5 ˚C and dry samples imaged by maintaining chamber pressure at 5 to 5.5 Torr. To encourage water condensation, the chamber pressure was increased to 6.7 Torr. Once the sample became fully submerged in water the surface was no longer visible so after allowing sample expansion to occur, the pressure was decreased again to 5.5 Torr. This caused water to evaporate and images of hydrated tissue were rapidly acquired.

For the mechanical stress experiment, pappus hairs were removed from apical plate samples using forceps and samples were submerged in water for at least 1 h. Samples were imaged using a dissecting microscope and then an incision was carefully carried out using a razor blade. Samples were briefly rehydrated and surface water removed with an absorbent tissue before imaging again. Samples were allowed to dehydrate for 10–20 min and imaged once again. Finally, samples were rehydrated and imaged one more time to check reversibility and that the incision had not damaged the structure too heavily.

**Raman spectroscopy.** Longitudinal 10 μm sections (see Microscopy and histology) were mounted on $CaF_2$ cover slips, submerged in distilled water, and imaged with a Renishaw InVia Raman microspectrometer using a 63× water immersion objective, with a numerical aperture of 0.9. A 785 nm laser beam was used for excitation at a laser power of 100% and exposure time of 10 s. Spectra were obtained from at least 20 points per region (floral podium, vasculature, cortex and sides) for each sample and 5 samples from different individual diaspores were used.

A rolling ball baseline correction was applied to all spectra with a baseline identification window of 60 and a smoothing window of 3. Intensity values were normalised to the $CaF_2$ substrate-derived peak at 321 $cm^{-1}$.

**Image analysis.** To measure pappus angle from the moisture chamber imaging experiments, image identity was blinded and images were randomised before measurement. Using Fiji, the angle between the outermost filaments (excluding those that were not in focus) was measured. Measurements of tissue sizes and lengths were carried out using line or polygon measurement tools in Fiji. Circularity was defined as:

$$\text{Circularity} = 4\pi \frac{\text{Area}}{\text{Perimeter}^2} \qquad (1)$$

such that a perfect circle will take a value of 1, and values closer to 0 indicating a more elongated shape.

Autofluorescence was quantified by measuring the mean grey value in small regions of the floral podium from confocal images. For each sample, fluorescence was normalized to the autofluoresence derived from the pappus hairs at pH 7.

For landmark displacement and relative expansion measurements, z-stacks of confocal autofluorescence images were used as described above. Maximum projections of each z-stack were obtained and images were stitched together using the Fiji plugin, Pairwise Stiching[36]. The polygon measurement tool was used to generate an outline of both dry and wet apical plates. Matched landmarks for dry and wet images of each sample were manually annotated. These consisted of salient features, such as cell corners, ridges and holes. For each sample, one landmark was selected close to the centre around the junction between the floral podium and central cavity to act as a reference point. Displacement of each landmark was calculated relative to this central point.

For local expansion rates, a Delaunay triangulation was mapped onto the landmarks of the dry state samples using R package deldir[37]. Triangles that were not fully enclosed by the overall outline of the dry sample were excluded. The area

of each triangle was calculated and normalized area change calculated by dividing the area of a triangle when wet by its area when dry. The data were smoothed by calculating the arithmetic mean of the normalized area change for each triangle and its adjoining neighbouring triangles. Principal orientations of strain were calculated for each triangle and annotated as a cross scaled to 3 times larger than the original values for improved visibility[38].

Regions of the apical plate were designated by manually outlining the floral podium, vasculature, side regions and central cavity on the original images using the Fiji polygon tool. Triangles in the dry state that overlapped at least 40% with any of the dry state regions were assigned to those regions and all others were designated as cortex triangles. Mean area changes were calculated for each region for each sample and these values used to generate boxplots and for statistical analysis.

For a very small number of triangles the nonuniform expansion process rearranged nearby landmarks relative to one another such that triangles then appeared to overlap in the wet state. As it is not physically possible for cells to actually overlap in a connected cellular structure, these triangles were omitted from the analysis and comparisons between regions.

Geometrical measurements of the apical plate were obtained from images and used as inputs for the computational model. Measurements of the sizes and geometry of regions were taken from the wet-state confocal images of apical plate half-sections using Fiji. For some parameters such as the vascular displacement ($d_{vasc}$) and holding angle ($\theta$), measurements were taken from consistent points on the dry and wet images and differences calculated. For region density measurements, 10 μm sections stained with ruthenium red gave consistent red staining across all cell types. These images were thresholded using the automatic colour thresholding in Fiji, converted to a binary image and the ratio of stained to unstained pixels used to calculate the density of cell wall material in manually selected rectangular areas of each region.

**Atomic Force Microscopy.** We used Atomic Force Microscopy (AFM) to assess mechanical properties of individual cell walls in dry sections of the apical plate. AFM experiments were performed using a stand-alone JPK Nanowizard III microscope, driven by JPK Nanowizard software version 6.1.181. Experiments were carried out in air at room temperature. RTESP-300 cantilevers (Bruker) with a nominal force constant of 40 N/m, and a nominal radius of 8 nm were used for all the measurements.

Calibration of the spring constant of the cantilever, was performed in air using the method proposed by Sader[39], by using the web application https://sadermethod.org. First, a power spectrum of the thermally excited cantilever is recorded and fit using a single harmonic oscillator model (SHO), by the contact-free calibration tool on JPK acquisition software. Then the resonance frequency and the Q factor of the cantilever (obtained from the fit) are uploaded to the web application, which provides a spring constant value accompanied by a 95% confidence interval, based on data uploaded by the community. A first estimation of the deflection sensitivity (the parameter allowing conversion between the raw reading of the photodiode to cantilever deflection) is obtained at the first run of the contact free calibration. Once the value of the spring constant is known, a second spectrum is acquired in the contact-based modality and after the fitting, this calibrated spring constant value is uploaded by the user into the interface, which modifies the deflection sensitivity accordingly. This approach does not require the acquisition of a force curve on a hard sample to measure the deflection sensitivity, which preserves the cantilever tip from damage, especially with stiff cantilevers such as the one used here.

When the same tip was used for several experiments on different days, the SNAP protocol[40] was followed for recalibrations. At first, the laser is positioned as closely as possible to its previous position (generally by seeking the same sum signal on the photodiode). A thermal tune is acquired and the deflection sensitivity value is corrected in order to obtain a spring constant as close as possible to its calibrated value. Then the new deflection sensitivity and the reference spring constant are set in the acquisition software.

The strategy for AFM force measurements was the following. Wax-embedded sections of 10 μm were mounted on a glass slide and wax not removed to ensure sections lay flat. Samples were first mapped using the Quantitative Imaging (QI) mode. Cell walls were localised using slope maps (displaying the slope of the last 50 nm of the extend curve) and adhesion maps (displaying the minimum force measured on the retract curve), based on both large modulus and small adhesion (see below). Then an average of 30 measurement positions were selected on cell walls; a force curve was recorded for each position. This procedure was adopted for each of the 4 regions of the sample. On several samples (6 out of 11), the wax embedding the slice was added as a control region. An effective elastic modulus was deduced for each measured point.

Scan size of QI maps was generally 40 × 40 μm² with a pixel size of 500 nm (80 × 80 pixels). A value of 1 μN for the trigger force was selected in order to obtain a clear contrast between cell walls and wax on slope maps and to make it easier to correctly place the measurement points. Z length (the ramp size of the force curve) was 2 μm; extension and retraction piezo speeds were 200 μm/s.

The force curves for the measurement of effective Young's modulus were made of 7 segments: 2 extension-retraction cycles (4 segments), a 3rd extend segment, a pause at constant force during 10 s, and a final retraction. The trigger force was

3 μN, the Z length was 2 μm (with 2000 pixels) and piezo speed (for both extend and retract) was 10 μm/s. The deduction of the effective Young's modulus was based on the last retraction segment. This load function (made of 7 segments) was defined in order to minimize the impact of viscous/plastic effects on the measurement of the elastic moduli. This strategy was motivated by the presence of a large hysteresis in the contact part of the force curves, between extension and retraction segments, which indicates a dissipation of energy (see for example ref. [41] and ref. [42]), as well as by the fact that the shape of the extension segment is more irregular than that of the retraction one (Supplementary Fig. 8).

Data analysis was performed with JPK Data Processing software 6.3.36. Force vs. height curves were first flattened by removing the result of a linear fit to the non-contact part of the force curve, in order to set it to 0 force. A first estimation of the point of contact (POC) was obtained by seeking for the first point crossing the 0 of forces, starting from the end of the first extend segment (i.e. trigger force position). The force vs. tip-sample distance was then obtained calculating a new axis of distances as height [m] – cantilever deflection Δd [m]. Young's modulus was obtained by fitting the last retraction segment of each curve with a Sneddon model[43] for a conical indenter, by using the following equation:

$$F = \frac{E}{1 - v^2} \frac{2\tan\alpha}{\pi} \delta^2 \qquad (2)$$

where $\alpha$ is the tip half angle, $\delta$ and $F$ indentation depth and applied load (both obtained from height, deflection, and cantilever stiffness), $v$ the Poisson's ratio and $E$ the Young's modulus. Although this model assumes no adhesion, perfectly elastic behaviour, homogeneous and isotropic sample, it provides a good fit to experimental curves and enables the comparison of different regions of the sample. For our analysis, we used an $\alpha$ of 18° and a $v$ of 0.5 (as is conventionally set for biological materials). The Young's modulus, the POC and an offset in force are free parameters of the fit.

In order to remove data coming from curves acquired on wax rather than on the sample, the force curves were filtered based on the adhesion force measured on the last retraction segment: curves acquired directly on wax systematically show higher adhesion values than on the exposed surface of the sample (see Fig. S8). A threshold was set to 160 nN on the adhesion, and all of the curves having an adhesion force below this value were kept. This empirical criterion has been double checked on the curves acquired on wax beside the sample, for which all adhesion forces were above this threshold. The median of Young's moduli was calculated for every region of each sample.

**Statistical analysis**. Normal distribution of data was confirmed using Shapiro-Wilk tests and significant differences were detected using ANOVAs and Tukey's honest significant difference test for differences in pappus angle and for all comparisons of quantitative data between regions of the apical plate. $\eta^2$ (Cohen's $d$) was calculated as a measure of effect size. For relative fluorescence at varying pH (Fig. 3d), data were not normally distributed according to a Shapiro-Wilk test so a paired Wilcoxon's signed rank test was used.

Nonlinear mixed effects models were fitted to the time courses of radial and longitudinal expansion using R package 'lme4'[44]. For both radial and longitudinal expansion, a negative exponential function was fitted using a modification of the 'SSAsymp' function in lme4. Time and measurement location were fixed effects while sample ID was a random effect, and intercepts were constrained to 0. Models were selected based on comparing Akaike information criteria (AIC). The optimal model for radial expansion had slopes (natural logarithm of the rate constant) and asymptotes varying according to sample ID and measurement location. This model performed better than versions in which the slope, asymptote or both only varied by sample and not measurement location. For longitudinal expansion, the optimal model had asymptotes varying by sample ID and measurement location, and slopes varying by sample ID. This performed better than similar models in which both asymptote and slope varied by sample ID and measurement location, or neither asymptote nor intercept varied by measurement location.

Correlation coefficients were calculated for geometrical parameters using the R package 'corrplot'[45] and the biplot of principal components plotted from a principal component analysis computed in R.

For all boxplots, the centre line is the median, hinges indicate first and third quartiles, and whiskers extend to largest value no further than 1.5 times the interquartile range.

**Modelling and supporting experimental data**. A detailed description of the model is included in the Supplementary Note 1 and a brief description provided here.

The hydrated state of the actuator corresponds to the geometry of the living tissue with turgor pressure removed. We consider that in this case, all stresses are relaxed. Therefore, an important assumption of our model is that the hydrated state of the actuator is stress-free. We tested this by hydrating apical plate samples and cutting part of the sample (Supplementary Fig. 9). In the hydrated state, incisions appeared quite closed with only a narrow gap forming between cut surfaces. Once dried, the incision gaped and a large opening was visible, which restored to a closed orientation when samples were rehydrated. This confirms that hydrated samples exhibit reduced mechanical stress.

Starting with this stress-free state, we modelled the observed longitudinal section of the actuator as a two-dimensional system in the planar strain condition, in which deformations are restricted to a plane. We chose this condition reasoning that it enabled stress in the direction orthogonal to the plane and would better reflect the three-dimensional nature of the actuator. We considered it an isotropic, linearly elastic material, which undergoes shrinkage due to loss of water. The corresponding partial differential equation corresponds to stress balance, with a source term associated with shrinkage. These equations are solved using FreeFem[46].

The starting hydrated geometry of the system is divided into 4 regions, with an arrangement and dimensions that follow experimental measurements. The displacement of vasculature at the basis of the actuator was prescribed according to experimental measurements. All regions have the same Poisson ratio. The podium, vasculature, and sides are characterised by the ratio of their modulus to that of cortex, which was inferred from density of cell wall material in experimental data. The Young's moduli, measured on individual cell walls by atomic force microscopy, were found to be similar between regions (Supplementary Fig. 8), so tissue-scale elasticity was based on cell wall density. We assumed that effective tissue elasticity is linearly proportional to the tissue density. This relationship has been demonstrated for woody materials that are stretched in all directions, as we expect to occur for a process involving swelling[47]. Each region has an intrinsic swelling parameter, which cannot be directly measured. Swelling parameters were determined by minimizing the difference in observed shrinking between model and experiments. We call the model, together with the measured and fitted parameters, the reference model.

The main prediction of the model is the holding angle, $\theta$. A higher value of $\theta$ indicates a more substantial pappus opening during dehydration indicating more efficient functioning of the actuator. Model sensitivity to a parameter was assessed from the derivatives of the angle $\theta$ with respect to this parameter around neighbouring values to the reference value. Correlations between $\theta$ and geometric parameters were predicted by combining the sensitivity values with the correlation matrix of geometric parameters (see Supplementary Note 1 for details).

**Reporting summary**. Further information on research design is available in the Nature Research Reporting Summary linked to this article.

## Data availability
Source data for this paper are available at the following repository: https://doi.org/10.5281/zenodo.6460887.

## Code availability
The implementation of the model as a FreeFem[46] script can be found in the following repository: https://doi.org/10.5281/zenodo.6424448.

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

## Acknowledgements

We thank Jim Buckman at Heriot-Watt University for use of the environmental scanning electron microscopy facility and Erika Kroll for assistance with pappus imaging. We also thank staff of the Nikolai Lobachevsky Library of Kazan Federal University for access to a scanned copy of the paper by Taliev in ref. [19]. and John Richards for aid in identifying the dandelion species section. This project was supported by a Leverhulme Trust grant (RPG-2015-255) to N.N. and I.M.V., a Scottish Universities Life Sciences Alliance Postdoctoral and Early Career Research Exchange grant to M.S., and a French National Research Agency grant (ANR-17-CE20-0023-02 "WALLMIME") to A.B. M.S. is supported by Leverhulme Trust Early Career Fellowship (ECF-2019-424); I.M.V. by a European Research Council grant (H2020 ERC-2020-COG 101001499); and N.N. by a Royal Society University Research Fellowship (UF140640 and URF\R\201035).

## Author contributions

M.S., A.K., E.M., A.B., and N.N. designed the experiments; M.S., A.K., and S.B. conducted the experiments and analyses; M.S., A.K., S.B., I.M.V., E.M., A.B., and N.N. contributed to interpretation of the results; M.S., A.K., A.B., and N.N. drafted the paper; and M.S., A.K., S.B., I.M.V., E.M., A.B., and N.N. revised the paper.

## Competing interests

The authors declare no competing interests.
