## [Peer Review File · Nature Communications]

Dandelion pappus morphing is actuated by radially patterned material swellingREVIEWER COMMENTS

Reviewer #1 (Remarks to the Author):

This manuscript studied the dandelion pappus morphing. To be honest, I don't know much about the relevant research field. However, the author used the Raman spectrum that I am familiar with to characterize ferulic acid and lignin. I have some questions about this part.

1. Line 677. Please provide the numerical aperture of the objective lens, so that we can estimate the spot size of Raman. If the spot size is very small, it may not be enough to obtain 20 points.
2. Line 685. There is no doubt that 1602 cm⁻¹ corresponding to lignin since it referred to aromatic ring of lignin. However, ferulic acid also has aromatic ring. Therefore, it is impossible to distinguish between lignin and ferulic acid in Raman spectra if the two components existing simultaneously. The peak of 1631 cm⁻¹ should be regarded as the drift of 1602 cm⁻¹ peak caused by the change of chemical environment. More evidence is needed to define it as ferulic acid.
3. The ratio of ferulic acid to lignin was calculated in this manuscript. How to calculate this ratio was not explained. However, whether the peak area or peak height is used, the correction curve should be made. The relationship between Raman characteristic peak and component content is unknown. In my opinion, if the Raman analysis is not important to this manuscript, this part could be deleted.

Reviewer #2 (Remarks to the Author):

Seale and colleagues have submitted a very interesting manuscript on the architecture and functional principle of the dandelion seed. The article is very well written and understandable, and the topic is of suitable interest for the Nature Communications readership. I have the following questions and concerns, which the authors should take into consideration during a revision:

Major points:

L570-579 "In the dandelion apical plate, we cannot rule out intrinsic swelling anisotropy as a possibility"

I agree with the authors that their model is indeed capable of explaining the general "macroscopic" deformation process, i.e. the way how the seed apparatus deforms and pappi are being displaced. However, it is nonetheless a pity that they cannot explain the "true biology" behind this phenomenon, as they state by themselves. With their model we have a nice understanding of the movement, but we are still unclear about the microscopical structural basis for this behaviour.

L610 Methods, Plant material

It is also a pity that the authors do not know the species they investigate. It is written in the methods that, initially and for a prior study, dandelion seeds were collected somewhere in Edinburgh. The authors indicate *Taraxacum officinale* as species (in their prior study it is *T. officinale* agg., which is a little more honest). However, the genus *Taraxacum* comprises several hundreds of species, which are notoriously difficult to identify. Have the authors tried to do so? Did they consult a specialist for identification? How sure are they regarding the epitheton "officinale" or the species group "agg."? Correct identification of the organism under investigation is crucial since the scientific literature is full of papers dealing with misidentified organisms, and this is very concerning! The paper under review here deals with a botanical structure, so it is important to be precise with taxonomical details. How sure are the authors that the results gained here with species x also apply for species y, z, and so on? Furthermore, was pollen introduction (by insects or other vectors) prevented during cultivation? By no means all *Taraxacum* species are apomictic, see Tas & van Dijk 1999 *Heredity* 83. And since the authors didn't know which species they had, I would be careful with the statement that all diaspores were genetically identical.

L625 onwards, methods:

The authors used methacrylate nail polish for blocking water uptake. I assume that the polish alters the mechanical properties of the structure and could, potentially, have blocked movement at all (or reduce it to a smaller extent, see my other comment). A good blocking layer could have been applied using Vaseline, which should not have a strong mechanical effect on the motion.

L757 onwards, methods: "Therefore, an important assumption of our model is that the hydrated state of the actuator is completely stress-free."

Could this not have been tested by cutting parts of the structure and observe potential gaping?

Minor points:

L45-46: "...to cause bending twisting."

"or" missing?

L76: "S. lepidophylla, cells that..."

Delete comma.

Figure 1 b and e: The pappi seem to move upwards a little (when the upper side of the plate is blocked) and downwards when the lower side is blocked. These minute effects might be caused by the mechanics of the nail polish, which may act as a resistance layer. See also my comment on the methods, regarding the suitability of the polish.

Figure 1g: A, B, and C are not explained in the legend, however, it is clear from Figs. 1a, b, d, and e what they mean. Perhaps the authors want to explain this additionally in the legend nonetheless.

L165: "As we had confirmed that the apical plate behaved as an actuator..."

It would have been nice if the authors had looked the very base of the pappi to rule out that there is a distinct and small bending zone.

L170-172: "above all of this, is a distinct layer of tissue that originally serves as a nectary and podium for the floral organs before floral abscission and cell death occurs to form the mature pappus."

This section could lead to misunderstanding, at least for me this sounds a bit strange since it is not clear how floral abscission and cell death leads to the formation of the "mature" pappus. Why mature at all? The pappi are the sepals of the individual flower, forming the calyx, and are not being formed with these processes.

L 181: I assume that altering the pressure inside the ESEM chamber does not affect the morphology of the biological specimen? Has this been tested?

L210: Figure 3c is mentioned before 3b. Please check consistence throughout manuscript.

L219-202: The color code does not have to be mentioned in the main text, it is sufficient that it is mentioned in the Fig. 3 legend.

Figure 5 and elsewhere: I did not completely understand how the pappus holding angle θ is defined, and it would have been nice to indicate this, for example, in Fig. 5a. I see in Line 310 that "The angle, θ , between the edges of the side region and the vertical in the dry state represents the pappus holding angle", but that does not really help much.

Generally, I would appreciate if the authors would not "only" present lateral views of the pappus actuator, but also a more inclined view to see in it "quasi-3D action". A nice supplementary video would be great.

Discussion: Could the authors (shortly?) comment on how the actual 3D deformation on the cellular level would look like? I assume that there are some crazy deformation going on with each cell; how are cells capable to do so?

L529: "since swelling regions must be connected to regions that do not swell".
Or ...to regions that swell lesser, or in another direction.

L544-556: "A 3D computational model may change this requirement as the remaining tissue would be radially connected."

As the authors state by themselves, a 3D model would have been (much?) more appropriate to verify the functional principle of the seed apparatus. Why was no such approach developed and computed?

L563-566: "Another possibility is that the lipid component maintains cells in a flexible but hydrophobic state such that they are able to expand and hold water within them. This would suggest that they act more like a balloon holding fluid inside than a sponge that absorbs water into its material structure." Such a behavior would be relatively easy to verify microscopically. Balloon cells should be only capable of hydraulic in- and deflation when their structural integrity is kept. Hygroscopic cell material swells and shrinks also when it is torn.

L620: what was the precision of the datalogger?

Reviewer #3 (Remarks to the Author):

It is very excellent paper in that the mechanism by which the opening angle of the dandelion's pappus changes is elucidated from multiple perspectives such as biology, materials science, chemistry, and mechanics by making full use of various experimental and computational techniques. Below are my questions and comments to the paper. I hope if I can contribute to improving the quality of this paper.

Questions / comments on the main text

01) P.3, L.118: It is better to show the definition of "the pappus angle" in the figure. This makes it clear the angle between which point to which point is called the pappus angle.

02) P.3, L.125 "partially prevented papas closure": The meaning is ambiguous. Does it mean that the prevention of pappus closure is inadequate?

03) P.4, L.126: "blocking the lower side (C) almost completely prevented closure" : As looking at the figure, it not only completely prevents closure, but it seems to be even more open. Is "blocking the lower side" more effective than preventing the closure of the pappus?

04) P.4, L.148 Figure 1.g: What is a, ab, and b in the figure indicate?

05) P.5, L.159 Figure S2: Is the pappus angle measurement accurate? If the target filament is not arranged in a plane perpendicular to the camera axis, an error will occur in the angle. Is the measurement error negligible?

06) P.5, L.159 Figure S2: From the graph, it seems that the plucked sample has a larger angle change than the intact sample. Is it necessary to consider the cause of the difference?

07) P.6, L.183 "the outlines of some cortical cells towards the lowermost corners of the bulging regions were visible (Fig 2c-d)": The outlines of some cortical cells are not clear in the image. Please show the places you can see with arrows etc. in the figure.

08) P.7, L.187 Figure 2. Why does the graph of h show the expansion rate, while the graph of j shows the change in length? Similar to the graph of h, if the expansion coefficient is shown in the range of 0% to 40% in the graph of j, it can be shown more clearly that the difference in the expansion coefficient between the upper part and the lower part is small.

09) P.9, L.224 Figure 3.g: What is a and b in the figure indicate?

10) P.10, L.248 "Using a central landmark close to the middle point of the floral podium as a reference, the relative displacement of landmarks was calculated (Fig 4e)." The displacement values will be changed depending on the reference point. What is the reason for using a central landmark close to the middle point of the floral podium as the reference point for displacement? Is it reasonable to use this as a reference?

11) P.12, L.288 "We considered the actuator as a two-dimensional, isotropic, linearly elastic system that undergoes shrinkage due to loss of water." Even in the 2-dimensional condition, there are a "planar stress condition" and a "planar strain condition". Please specify which condition was used for the 2-dimensional model of the apical plate.

12) P.12, L.300, Figure 5. g and h: "tensile stress" should be "principal tensile stress". "compressive stress" should be "principal compressive stress". Value of tensile stress or compressive stress depend on the direction. The principal stress is the stress in the direction in which the stress is maximum or minimum.

13) P.13, L.309 "the base of the vascular bundles displaced by a fixed amount (Table 1).": Table 1 shows the value of d_{vasc} is $14.6 \mu\text{m}$. In the supplementary note, the value of d_{vasc} is shown as $15.3 \mu\text{m}$ on the section of "the Boundary condition. Which is correct?

14) P.13, L.310 " θ , between the edges of the side region and the vertical in the dry state": The edges of the side region is a curve, so is it okay to understand that the angle θ is the tangent angle to vertical lines at the corner? Although it is written in the method, it is better to specify the θ is a tangent angle mathematically. In addition, the definition of θ should be shown in figure 5 as well.

15) P.13, L.325 "a Poisson ratio typical for woody tissues": Please show the literature as a reference specified the value of Poisson's ratio.

16) P.15, L. 357 "the reference model sufficiently recapitulates the observed behavior of the pappus actuator.": It is necessary to show that the recapitulate is qualitative rather than quantitative.

17) P.18, L. 472 Figure 6. a-b: It is better to explain that the sensitivity is normalized as a dimensionless quantity and the numerical value on the vertical axis of the graph is a dimensionless quantity.

18) P.22, L.512 "This arrangement allows precise radial swelling that is not seen in other hygroscopic movements.": It seems exaggeration to mention three-dimensional radial swelling from a two-dimensional study. Since the radial swelling is related to the axially symmetric shape of the apical plate and its deformation, it is necessary to consider the axial symmetry.

19) P.22, L.544 "A 3D computational model may change this requirement as the remaining tissue would be compete connected.": It is necessary to mention the axisymmetric computational model before the 3D computational model. In the axisymmetric model, it is possible to analyze an axisymmetric solid in two dimensions by assuming that the strain in the circumferential direction is zero. (For example, Chien Wei-zang, Finite element analysis of axisymmetric elastic body problems, Applied Mathematics and Mechanics volume 1, pages 23–34 (1980))

In your paper, the analysis is performed assuming the expansion and contraction of the apical plate as a two-dimensional problem, but it is more appropriate to treat it as an axisymmetric problem. When considered as an axisymmetric model, the volume of the outer cortex and side region is larger than that of the two-dimensional model, so it is expected that the strain and stress generated by swelling will also be larger. Therefore, it is considered that the axially symmetric model has a larger deformation of the upper side surface and a larger change in the opening pappus angle. To carry out the axisymmetric analysis is required in the next research step, but in this paper, it is necessary to consider whether it is more consistent with the experiment when considered as an axisymmetric model.

20) P.23, L.573, "we find from our computational model that intrinsic anisotropy is not necessary for a functional model." :

Since vasculature has a large anisotropy with respect to the fiber direction, it is expected that the influence on the expansion / contraction deformation of the entire apical plate is not small. Is it possible that the tendency shown in Figures S5 and S6 will change when the intrinsic anisotropy in the material is taken into account?

Questions / comments regarding the supplementary material "The mechanism of dandelion pappus closure Supplementary note"

21) P.2, Elastic parameters "the relative Young's moduli are approximated by relative densities of regions (quantified as the volume of cell wall per unit area)":

This method assumes that the Young's modulus is proportional to the density. It is necessary to show the rationale for such a relationship. Also, in the calculation, it is necessary to show at least the Young's modulus value of the vasculature which is the reference for others.

22) P.3, Table 2.: Table captions should be laid out above the table. Only Table 2 is shown in the supplementary material, but does Table 1 not exist?

REVIEWER COMMENTS

Reviewer #1 (Remarks to the Author):

This manuscript studied the dandelion pappus morphing. To be honest, I don't know much about the relevant research field. However, the author used the Raman spectrum that I am familiar with to characterize ferulic acid and lignin. I have some questions about this part.

1. Line 677. Please provide the numerical aperture of the objective lens, so that we can estimate the spot size of Raman. If the spot size is very small, it may not be enough to obtain 20 points.

The numerical aperture was 0.9, which we have now added to the manuscript (line 759). This would give an approximate spot size of 1 μm , which is considerably larger than the expected size of cell wall polysaccharides.

2. Line 685. There is no doubt that 1602 cm^{-1} corresponding to lignin since it referred to aromatic ring of lignin. However, ferulic acid also has aromatic ring. Therefore, it is impossible to distinguish between lignin and ferulic acid in Raman spectra if the two components existing simultaneously. The peak of 1631 cm^{-1} should be regarded as the drift of 1602 cm^{-1} peak caused by the change of chemical environment. More evidence is needed to define it as ferulic acid.

Thank you for pointing this out. We carried out the Raman spectroscopy to gain further insight into the components comprising the cell walls of different regions. In quantifying the ferulic acid to lignin ratio, we used the method described in a previous publication (Abraham, Y., Dong, Y., Aharoni, A. et al. *Mapping of cell wall aromatic moieties and their effect on hygroscopic movement in the awns of stork's bill. Cellulose* 25, 3827–3841 (2018). <https://doi.org/10.1007/s10570-018-1852-x>). This analysis was related to specific peaks for ferulic acid observed in another publication (O. Piot, J.-C. Autran, M. Manfait, *Investigation by Confocal Raman Microspectroscopy of the Molecular Factors Responsible for Grain Cohesion in the Triticum aestivum Bread Wheat. Role of the Cell Walls in the Starchy Endosperm, Journal of Cereal Science*, 34. 191-205 (2001) <https://doi.org/10.1006/jcrs.2001.0391>). The presence of a unique peak in the Raman spectrum of the floral podium tissue similar to the peaks observed in Abraham et al (2018) and Piot et al (2001) was therefore indicative of ferulic acid. Furthermore, this was supported by our histological experiments that demonstrated limited lignin (phloroglucinol) staining in the floral podium but high autofluorescence under UV light (consistent with ferulic acid presence).

While our multiple lines of evidence indicated the presence of ferulic acid but limited lignin in the floral podium, we acknowledge the reviewer's concern about the overlap in spectra between these compounds and the issues with attempting to quantify particular components. As a result, we have modified the text we use to describe the Raman spectroscopy results to focus on a more qualitative framing of the data, and we have added in a citation to the Piot (2001) paper to justify our comments more appropriately (lines 246-249). We have removed the quantification of lignin/ferulic acid ratio from the manuscript to avoid any confusion.

3. The ratio of ferulic acid to lignin was calculated in this manuscript. How to calculate this ratio was not explained. However, whether the peak area or peak height is used, the correction curve should be made. The relationship between Raman characteristic peak and component content is unknown. In my opinion, if the Raman analysis is not important to this manuscript, this part could be deleted.

The peak height was used for the quantification but this is now not relevant as, based on the comment above, we decided to remove this analysis from the manuscript. We have now focused on qualitative descriptions of the Raman data and explaining how this fits together with the histological data we also describe.

.....
Reviewer #2 (Remarks to the Author):

Seale and colleagues have submitted a very interesting manuscript on the architecture and functional principle of the dandelion seed. The article is very well written and understandable, and the topic is of suitable interest for the Nature Communications readership. I have the following questions and concerns, which the authors should take into consideration during a revision:

Thank you for the encouraging response and for your careful review. We address the individual points below.

Major points:

L570-579 “In the dandelion apical plate, we cannot rule out intrinsic swelling anisotropy as a possibility”

I agree with the authors that their model is indeed capable of explaining the general “macroscopic” deformation process, i.e. the way how the seed apparatus deforms and pappi are being displaced. However, it is nonetheless a pity that they cannot explain the “true biology” behind this phenomenon, as they state by themselves. With their model we have a nice understanding of the movement, but we are still unclear about the microscopical structural basis for this behaviour.

We agree that we have explained the deformation process to a considerable degree as we have established a framework of expansion/contraction for specific regions of tissue, aspects of their chemical composition, and the way in which each part contributes to the overall deformation of the structure. In terms of the underlying biology, we have also demonstrated through our modelling that the differences in tissue density (and therefore the size of the cells) and the geometrical arrangement of the different regions are major contributors to the actuator behaviour we observe.

Although there may be additional features involved that we have not fully explored, we find that the aspects of the mechanism that we have uncovered are largely sufficient to explain the phenomenon. In this way, we have established the fundamental biological patterning that allows the apical plate to function. Indeed, we were planned to characterise the cellular structural drivers for anisotropic expansion (e.g. cellulose orientation). However, after the model revealed that such cellular structural features were not necessary for the actuator function, we decided not to pursue such experiments, as the result would not have been critical either way.

L610 Methods, Plant material

It is also a pity that the authors do not know the species they investigate. It is written in the methods that, initially and for a prior study, dandelion seeds were collected somewhere in Edinburgh. The authors indicate *Taraxacum officinale* as species (in their prior study it is *T. officinale* agg., which is a little more honest). However, the genus *Taraxacum* comprises several hundreds of species, which are notoriously difficult to identify. Have the authors tried to do so? Did they consult a specialist for identification? How sure are they regarding the epitheton “*officinale*” or the species group “agg.”? Correct identification of the organism under investigation is crucial since the scientific literature is full of papers dealing with misidentified organisms, and this is very concerning! The paper under review here deals with a botanical structure, so it is important to be precise with taxonomical details. How sure are the authors that the results gained here with species x also apply for species y, z, and so on?

Furthermore, was pollen introduction (by insects or other vectors) prevented during cultivation? By no means all *Taraxacum* species are apomictic, see Tas & van Dijk 1999 *Heredity* 83. And since the authors didn't know which species they had, I would be careful with the statement that all diaspores were genetically identical.

We appreciate the reviewer's point here and have made some effort to address this.

We consulted John Richards, the foremost expert on UK dandelions who informed us from images of our plants that the species we used was from the section 'Taraxacum' (formerly known as Ruderalia) and that this would certainly be a triploid apomict. Unfortunately, due to the greenhouse growth conditions of our plants, he was unable to identify the species more specifically though he thought it likely to be *T. caloschistum*. He could not confirm this without observing plants grown in more natural conditions. To achieve this, we would need to sow seeds outdoors now that would only flower in Spring 2023, which is well beyond the timeline of this revision.

We understood the reviewer's concern is that the samples examined in this work are not specified enough, and this we can address thoroughly. All samples used were from the same plant collected from a specific meadow located on the King's Building campus of the University of Edinburgh (55.922502, -3.170236). We have clarified this in the methods (line 682-3). Additionally, while the quantitative aspects of the data may well vary between other dandelion species, we expect that the qualitative behaviour would occur, given that many Asteraceae beyond dandelions carry out this type of morphing. To reduce confusion, we have relabelled the species as '*Taraxacum officinale* agg.' as previously recorded (line 685-8).

L625 onwards, methods:

The authors used methacrylate nail polish for blocking water uptake. I assume that the polish alters the mechanical properties of the structure and could, potentially, have blocked movement at all (or reduce it to a smaller extent, see my other comment). A good blocking layer could have been applied using Vaseline, which should not have a strong mechanical effect on the motion.

The methacrylate nail polish was deliberately used to block the mechanical motion of the apical plate. The description in the main text reflected our wish to point out that there may have also been a possible secondary effect on water uptake. Blocking the mechanical action tested whether the apical plate motion is required for pappus morphing (in contrast to motion of the hairs themselves for example). Although the sites of water uptake are also relevant and interesting, that would be more likely to affect the dynamics of pappus morphing rather than the steady state behaviour (unless all water uptake is prevented).

L757 onwards, methods: "Therefore, an important assumption of our model is that the hydrated state of the actuator is completely stress-free."

Could this not have been tested by cutting parts of the structure and observe potential gaping?

This is a great suggestion and we have carried out an additional experiment to test this assumption. We have made some incisions in the structure and observed significant gaping in the dry apical plate samples but minimal gaping in the hydrated samples. We have included images of this as Figure S9, and have added some descriptions of this to the text (lines 388-391, 939-943).

Minor points:

L45-46: "...to cause bending twisting."

"or" missing?

Thank you for noticing this error – we have added the missing word (line 57).

L76: "S. lepidophylla, cells that..."

Delete comma.

This is now corrected (line 88).

Figure 1 b and e: The pappi seem to move upwards a little (when the upper side of the plate is blocked) and downwards when the lower side is blocked. These minute effects might be caused by the mechanics of the nail polish, which may act as a resistance layer. See also my comment on the methods, regarding the suitability of the polish.

For Figure 1b where nail polish is applied to the upper side of the apical plate, it is true that the pappus filaments raise upwards slightly. This is reflected in the quantitative data and indicates that the pappus morphing is can partially close but that motion is somewhat prevented by applying the nail polish to the upper side. The effect is likely to be due to the mechanical resistance provided by the nail polish to the upper side. The lower side is still able to expand (at least to some extent) though leading to partial pappus morphing.

For 1e, it does appear that the pappus is slightly more open (some hairs point slightly downwards) compared to 1a. This is likely to be because these images are of different individual samples so exhibit small variations in pappus morphology. Different samples were used for each treatment as nail polish could not be removed after it was applied. We have added a few words to the figure legend to make this clear (line 159-160).

Figure 1g: A, B, and C are not explained in the legend, however, it is clear from Figs. 1a, b, d, and e what they mean. Perhaps the authors want to explain this additionally in the legend nonetheless.

Thank you for pointing this out. We have added a full explanation in the figure legend (line 164-165).

L165: "As we had confirmed that the apical plate behaved as an actuator..."

It would have been nice if the authors had looked the very base of the pappi to rule out that there is a distinct and small bending zone.

This is an excellent point. We did in fact also image some of the pappus hairs that had been detached from the apical plate in the environmental SEM. When the chamber pressure was altered to change the level of water condensation on the hairs, we did not observe any change in size or motion of the hairs themselves (see images below). In Supplementary Video 1 already provided, it is also possible to observe the base of the hairs where they are attached to the apical plate. In this video it is also clear that the hair cells do not change in size or shape.

Above images: ESEM of pappus hairs dry (left) and wet (right).

L170-172: “above all of this, is a distinct layer of tissue that originally serves as a nectary and podium for the floral organs before floral abscission and cell death occurs to form the mature pappus.” This section could lead to misunderstanding, at least for me this sounds a bit strange since it is not clear how floral abscission and cell death leads to the formation of the “mature” pappus. Why mature at all? The pappi are the sepals of the individual flower, forming the calyx, and are not being formed with these processes.

We see that this sentence was confusing and have altered the text (line 190-3) to read as follows:

‘Situating above all of this, is a distinct layer of tissue that originally serves as a nectary and podium for the floral organs before floral abscission and cell death occurs. The dead calyx tissue remains behind and is thereafter named the pappus.’

L 181: I assume that altering the pressure inside the ESEM chamber does not affect the morphology of the biological specimen? Has this been tested?

We are confident that the pressure changes do not alter the morphology of the specimen. Firstly, the pressure differences were kept as close as possible to the condensation/evaporation point for water to minimise changes in pressure. Secondly, no instantaneous changes in morphology were observed when the pressure changed, but only after water droplets began to visibly form on the sample. Thirdly, if the pressure was affecting the morphology, we would expect it to act in the opposite direction to the observed morphing. A higher pressure might be expected to contract the specimen, but instead the sample expands (once water droplets are present).

Finally, we observe similar behaviour in both the ESEM images (Figure 2a-d) and the light microscope images (2e-f). In any case, all quantitative measurements used for the computational model were carried out on images from either the light microscope or confocal microscope, neither of which involve any pressure modifications.

L210: Figure 3c is mentioned before 3b. Please check consistency throughout manuscript. We have altered Figure 3 to match the order in which figures are mentioned through the manuscript.

L219-202: The color code does not have to be mentioned in the main text, it is sufficient that it is mentioned in the Fig. 3 legend.

Thank you for pointing this out. We have removed the duplication.

Figure 5 and elsewhere: I did not completely understand how the pappus holding angle θ is defined, and it would have been nice to indicate this, for example, in Fig. 5a. I see in Line 310 that “The angle, θ , between the edges of the side region and the vertical in the dry state represents the pappus holding angle”, but that does not really help much.

We have added in illustrations to Figure 1h and 5e to clarify how this angle is defined as well as describing this angle more clearly in the text (line 332-3, 345-7).

Generally, I would appreciate if the authors would not “only” present lateral views of the pappus actuator, but also a more inclined view to see in it “quasi-3D action”. A nice supplementary video would be great.

We have added in Supplementary Video 2, which is a top-down view of the pappus as it becomes hydrated in the moisture chamber.

Discussion: Could the authors (shortly?) comment on how the actual 3D deformation on the cellular level would look like? I assume that there are some crazy deformation going on with each cell; how are cells capable to do so?

This is a good point. We have added in an additional supplementary figure illustrating some of the unfolding of cortex tissue during hydration, and the resulting size and shape changes that occur. We have mentioned this in the discussion as suggested (line 205-6, 604-7).

L529: "since swelling regions must be connected to regions that do not swell".
Or ...to regions that swell lesser, or in another direction.

Thank you for this point – we have modified the text as suggested (line 576-7).

L544-556: "A 3D computational model may change this requirement as the remaining tissue would be radially connected."

As the authors state by themselves, a 3D model would have been (much?) more appropriate to verify the functional principle of the seed apparatus. Why was no such approach developed and computed?

While a 3D model would obviously be valuable, our experimental observations were all in 2D. For consistency, we therefore used a 2D modelling approach so that experimental and modelling observations could be more easily compared and optimised. Nevertheless, we have verified that an axisymmetric model appears to show very similar results to our 2D model (see comments in response to Reviewer 3).

L563-566: "Another possibility is that the lipid component maintains cells in a flexible but hydrophobic state such that they are able to expand and hold water within them. This would suggest that they act more like a balloon holding fluid inside than a sponge that absorbs water into its material structure."

Such a behavior would be relatively easy to verify microscopically. Balloon cells should be only capable of hydraulic in- and deflation when their structural integrity is kept. Hygroscopic cell material swells and shrinks also when it is torn.

Due to the extremely small size of the tissues involved, it would be experimentally unfeasible to isolate such small regions with the tools we have available. As this was a speculative point in the discussion that we cannot experimentally verify, we have removed this statement from the discussion.

L620: what was the precision of the datalogger?

We have added this information into the methods section (line 693-4).

.....
Reviewer #3 (Remarks to the Author):

It is very excellent paper in that the mechanism by which the opening angle of the dandelion's pappus changes is elucidated from multiple perspectives such as biology, materials science, chemistry, and mechanics by making full use of various experimental and computational techniques. Below are my questions and comments to the paper. I hope if I can contribute to improving the quality of this paper.

We appreciate the positive comments and have attempted to address all the issues raised as outlined below.

Questions / comments on the main text

01) P.3, L.118: It is better to show the definition of "the pappus angle" in the figure. This makes it clear the angle between which point to which point is called the pappus angle.

Thank you for this suggestion. We have now included diagrams of both the pappus angle and the holding angle in Figure 1 and Figure 5.

02) P.3, L.125 "partially prevented papas closure": The meaning is ambiguous. Does it mean that the prevention of pappus closure is inadequate?

We have added some extra detail to clarify this (line 137-9). In this condition, the pappus angle changes when the sample is hydrated but not to the same degree as the control samples.

03) P.4, L.126: "blocking the lower side (C) almost completely prevented closure" : As looking at the figure, it not only completely prevents closure, but it seems to be even more open. Is "blocking the lower side" more effective than preventing the closure of the pappus?

This appearance is due to the use of different samples for each image that exhibit slight variation in morphology (see comment to Reviewer 2). We have clarified that the images are of different samples in the figure legend to avoid confusion (line 159-60).

04) P.4, L.148 Figure 1.g: What is a, ab, and b in the figure indicate?

The letters above bars indicate statistically significant differences between groups, which we have clarified in the figure legend.

05) P.5, L.159 Figure S2: Is the pappus angle measurement accurate? If the target filament is not arranged in a plane perpendicular to the camera axis, an error will occur in the angle. Is the measurement error negligible?

The samples were set up as carefully as possible to ensure that the pappus was oriented perpendicular to the camera. For the angle measurements, hairs at the edges of the pappus were selected that were most in focus. Though there is likely to be a small amount of error in these measurements, it will be random across samples meaning that comparisons between groups should be valid.

06) P.5, L.159 Figure S2: From the graph, it seems that the plucked sample has a larger angle change than the intact sample. Is it necessary to consider the cause of the difference?

This was explained in the main text (line 151-153) 'This may be because clusters of hairs normally slightly obstruct one another during motion and removing hairs reduces this effect.'

07) P.6, L.183 "the outlines of some cortical cells towards the lowermost corners of the bulging regions were visible (Fig 2c-d)": The outlines of some cortical cells are not clear in the image. Please show the places you can see with arrows etc. in the figure.

Thanks for pointing this out. In response to this and to a point from Reviewer 2 we have added in a new supplementary Figure 3 illustrating the outline of the cortex cells and quantification of their size and shape changes.

08) P.7, L.187 Figure 2. Why does the graph of h show the expansion rate, while the graph of j shows the change in length? Similar to the graph of h, if the expansion coefficient is shown in the range of 0% to 40% in the graph of j, it can be shown more clearly that the difference in the expansion coefficient between the upper part and the lower part is small.

This is a good point – we have changed j to match h so that it is easier to make this comparison.

09) P.9, L.224 Figure 3.g: What is an and b in the figure indicate?

As before, these letters indicate statistically significant differences between groups, which we have clarified in the figure legend.

10) P.10, L.248 "Using a central landmark close to the middle point of the floral podium as a reference, the relative displacement of landmarks was calculated (Fig 4e)." The displacement values will be changed depending on the reference point. What is the reason for using a central landmark close to the middle point of the floral podium as the reference point for displacement? Is it reasonable to use this as a reference?

The displacement was mapped for illustrative purposes to indicate the direction of movement and the data were not used for any quantitative comparisons. The reference point is arbitrary so a central point was selected to reflect the fact that the structure is approximately radially symmetrical.

11) P.12, L.288 "We considered the actuator as a two-dimensional, isotropic, linearly elastic system that undergoes shrinkage due to loss of water." Even in the 2-dimensional condition, there are a "planar stress condition" and a "planar strain condition". Please specify which condition was used for the 2-dimensional model of the apical plate.

The planar strain condition was used and we have added this information to the manuscript (line 944-6).

12) P.12, L.300, Figure 5. g and h: "tensile stress" should be "principal tensile stress". "compressive stress" should be "principal compressive stress". Value of tensile stress or compressive stress depend on the direction. The principal stress is the stress in the direction in which the stress is maximum or minimum.

Thank you for pointing this out. We have modified the figure legend accordingly.

13) P.13, L.309 "the base of the vascular bundles displaced by a fixed amount (Table 1).": Table 1 shows the value of d_vasc is 14.6 μm . In the supplementary note, the value of d_vasc is shown as 15.3 μm on the section of "the Boundary condition. Which is correct?

We apologise for this error (and in fact some other minor errors in this table), which we have now fixed.

14) P.13, L.310 " θ , between the edges of the side region and the vertical in the dry state": The edges of the side region is a curve, so is it okay to understand that the angle θ is the tangent angle to vertical lines at the corner? Although it is written in the method, it is better to specify the θ is a tangent angle mathematically. In addition, the definition of θ should be shown in figure 5 as well.

Thank you for pointing out the possible confusion about the definition of the holding angle. Indeed, the edges of the side region are curved, and in order to approximate an experimentally measured 'holding angle', we considered taking the tangent angle (the angle between the vertical and the tangent vector to the edge curve) in the upper corner or the tangent angle in the lower corner of the

side edge. However, the pappus hairs are attached not only on one corner of the side region, but all along the side edge. The tangent angle in the upper corner would overestimate, while the tangent angle on the lower corner of the side region would underestimate the "average angle" on the edges of the side region. This is why, instead of choosing a tangent angle we have chosen as holding angle "the angle between the vertical and the line between the upper corner of the apical plate and the lowest point on the lateral edge of the side region". We have clarified the definition of the holding angle in the text, and included an illustration of theta in Figure 5e (line 345-7).

15) P.13, L.325 "a Poisson ratio typical for woody tissues": Please show the literature as a reference specified the value of Poisson's ratio.

Thanks for pointing this out, we have added in the reference (line 362).

16) P.15, L. 357 "the reference model sufficiently recapitulates the observed behavior of the pappus actuator.": It is necessary to show that the recapitulate is qualitative rather than quantitative.

We have clarified the sentence prior to this to demonstrate the similarities between the model in both quantitative and qualitative manners (line 396-8). Although our model does not perfectly match the experimental data quantitatively, our sensitivity analysis shows that the behaviour is robust and we have a degree of motion that would be sufficient to make a substantial difference to pappus functionality during flight, as we mentioned in the discussion (line 623-4).

17) P.18, L. 472 Figure 6. a-b: It is better to explain that the sensitivity is normalized as a dimensionless quantity and the numerical value on the vertical axis of the graph is a dimensionless quantity.

Thank you for this point, we have clarified in the text that the sensitivity values are non-dimensional and that values are therefore comparable with one another (line 409-11).

18) P.22, L.512 "This arrangement allows precise radial swelling that is not seen in other hygroscopic movements.": It seems exaggeration to mention three-dimensional radial swelling from a two-dimensional study. Since the radial swelling is related to the axially symmetric shape of the apical plate and its deformation, it is necessary to consider the axial symmetry.

We agree that the actuator is axisymmetric with a good approximation, and indeed, an axisymmetric three-dimensional model would be a next step in order to further deepen the study. However, given that the vascular bundles are discontinuous but evenly distributed around the symmetry axis, the structure has in fact a discrete axial symmetry. Therefore, we deliberately used the term 'radial symmetry' instead. Nevertheless, we did check the impact of axisymmetric modelling (see response to point 19).

19) P.22, L.544 "A 3D computational model may change this requirement as the remaining tissue would be completely connected.": It is necessary to mention the axisymmetric computational model before the 3D computational model. In the axisymmetric model, it is possible to analyze an axisymmetric solid in two dimensions by assuming that the strain in the circumferential direction is zero. (For example, Chien Wei-zang, Finite element analysis of axisymmetric elastic body problems, Applied Mathematics and Mechanics volume 1, pages 23–34 (1980))

In your paper, the analysis is performed assuming the expansion and contraction of the apical plate as a two-dimensional problem, but it is more appropriate to treat it as an axisymmetric problem. When considered as an axisymmetric model, the volume of the outer cortex and side region is larger than that of the two-dimensional model, so it is expected that the strain and stress generated by swelling will also be larger. Therefore, it is considered that the axially symmetric model has a larger

deformation of the upper side surface and a larger change in the opening pappus angle. To carry out the axisymmetric analysis is required in the next research step, but in this paper, it is necessary to consider whether it is more consistent with the experiment when considered as an axisymmetric model.

As suggested, we constructed and implemented the corresponding axisymmetric 3D model. The above figure shows in black the original undeformed mesh corresponding to the hydrated state of the actuator. The red mesh is the deformed (dry) state given by the 2D reference model presented in the article, where the measured horizontal displacement, d_{vasc} , was prescribed for the bottom of the vasculature. And finally, the blue mesh is the deformed (dry) state given as the result of the 3D axisymmetric model where no boundary condition is applied, and all parameters (geometrical as well as those related to material properties) are taken the same as in the 2D reference model of the article. Although we find that indeed, the side region has higher area change in the axisymmetric case than in the 2D case, we also find that the difference between the two deformed geometries and in particular in the predicted side angle values is very small (see table below).

All area change measurements presented in the article were done in 2D, on physically cut sections of the actuator tissue, and the difference in outcome between the 2D and the axisymmetric 3D model turns out to be below measurement errors. Therefore, we think that the 2D model is more appropriate to the experimental results presented in the article.

	Area change ($A_{hydrated}/A_{dry}$)				Holding angle (degrees)
	cortex	floral podium	side	vasculature	
2D	1.67	1.61	1.79	1.28	19.46
Axisymmetric 3D	1.71	1.62	1.93	1.24	19.48

20) P.23, L.573, "we find from our computational model that intrinsic anisotropy is not necessary for a functional model." :

Since vasculature has a large anisotropy with respect to the fiber direction, it is expected that the influence on the expansion / contraction deformation of the entire apical plate is not small. Is it possible that the tendency shown in Figures S5 and S6 will change when the intrinsic anisotropy in the material is taken into account?

Thank you for this interesting point. We agree that anisotropy of some cell types is quite likely and would probably have an effect on the deformation. If intrinsic anisotropy is present, we would expect it to enhance the deformation further so our results represent a lower bound on the magnitude of angle changes. Our results show that anisotropy is not inherently required for the morphing behaviour we observe as we explain in the discussion (line 615-21). While additional complexity may be present in the tissue, it is not required for the baseline level of deformation. Uncovering these additional subtleties will be the focus of our future investigations.

Questions / comments regarding the supplementary material "The mechanism of dandelion pappus closure Supplementary note"

21) P.2, Elastic parameters "the relative Young's moduli are approximated by relative densities of regions (quantified as the volume of cell wall per unit area)":

This method assumes that the Young's modulus is proportional to the density. It is necessary to show the rationale for such a relationship. Also, in the calculation, it is necessary to show at least the Young's modulus value of the vasculature which is the reference for others.

Thank you for this helpful point. We have added in a new Supplementary Figure 8 showing measurements of cell wall Young's moduli for all regions and an explanation of the relationship between cell wall modulus, density, and tissue-scale modulus (line 363-5, 954-9).

22) P.3, Table 2.: Table captions should be laid out above the table. Only Table 2 is shown in the supplementary material, but does Table 1 not exist?

Apologies for the confusion, we have relabelled this table as Supplementary Table 1.

REVIEWERS' COMMENTS

Reviewer #1 (Remarks to the Author):

The authors have taken into consideration the comments and recommendations of the reviewer and significant improvement changes have been implemented. Concerning the last version of the manuscript I could state that in my opinion the manuscript can be accepted in the present form.

Reviewer #2 (Remarks to the Author):

The authors did a great job in revising their ms. My sincere congratulations and thank you for clarification! The only remaining point or recommendation, probably rather for future studies, is that the authors should prepare a herbarium specimen and deposit in an open repository, so that taxonomical questions can be addressed. Apart from that, the paper can be published as it stands.

Reviewer #3 (Remarks to the Author):

I appreciate to the author's polite response to any of my questions and comments and for the appropriate revision of the paper. I received generally satisfactory answers, but I have some additional questions and suggestions for corrections, which are listed below. Please understand that I wrote additional questions & comments below the author's answer to the first question & comment.

01) P.3, L.118: It is better to show the definition of "the pappus angle" in the figure. This makes it clear the angle between which point to which point is called the pappus angle.

Thank you for this suggestion. We have now included diagrams of both the pappus angle and the holding angle in Figure 1 and Figure 5.

This revision makes the definition of "pappus angle and the holding angle" easier to understand. However, if the "holding angle" written on the apical plate in Fig.1h is set to "holding angle (θ)", θ in Fig.1 and Fig.5 can be shown more clearly.

03) P.4, L.126: "blocking the lower side (C) almost completely prevented closure ": As looking at the figure, it not only completely prevents closure, but it seems to be even more open. Is "blocking the" lower side "more effective than preventing the closure of the pappus?"

This appearance is due to the use of different samples for each image that exhibit slight variation in morphology (see comment to Reviewer 2). We have clarified that the images are of different samples in the figure legend to avoid confusion (line 159-60).

With this revision (line 165-166), the meaning of Figure 1g is better understood. However, in the main text shows "(A), (B), (C)", the description in Figure 1g and (line 165-166) shows "A, B, C". I think it is better to unify the description (A), (B), (C).

06) P.5, L.159 Figure S2: From the graph, it seems that the plucked sample has a larger angle change than the intact sample. Is it necessary to consider the cause of the difference?

This was explained in the main text (line 151-153)'This may be because clusters of hairs normally slightly obstruct one another during motion and removing hairs reduces this effect.'

I would like to comment that I was able to understand it well with the explanation of a valid reason.

07) P.6, L.183 "the outlines of some cortical cells towards the lowermost corners of the bulging regions were visible (Fig 2c-d)": The outlines of some cortical cells are not clear in the image. Please show the places you can see with arrows etc. in the figure.

Thanks for pointing this out. In response to this and to a point from Reviewer 2 we have added in a new supplementary Figure 3 illustration the outline of the cortex cells and quantification of their size and shape changes.

With new Figure S3, it is now possible to better understand the changes in cell shape. However, the characters in the S3c and S3d graph may be small and difficult to read.

10) P.10, L.248 "Using a central landmark close to the middle point of the floral podium as a reference, the relative displacement of landmarks was calculated (Fig 4e)." The displacement values will be changed depending on What is the reason for using a central landmark close to the middle point of the floral podium as the reference point for displacement? Is it reasonable to use this as a reference?

The reference point is arbitrary so a central point was selected to reflect the fact that the structure is approximately symmetrically.

In the above explanation, the reason for the reference point of displacement is understood. I think it's a good idea to add the explanation "The central point was selected to reflect the fact that the structure is approximately symmetrically." In the main text.

11) P.12, L.288 "We considered the actuator as a two-dimensional, isotropic, linearly elastic system that undergoes shrinkage due to loss of water." Even in the 2-dimensional condition, there are a "planar stress condition" Please specify which condition was used for the 2-dimensional model of the apical plate.

The planar strain condition was used, and we have added this information to the manuscript (line 944-6).

I would like to comment that "the planar strain condition" is considered to be an appropriate two-dimensional approximation condition to obtain the deformation of the holding angle.

16) P.15, L. 357 "the reference model sufficiently recapitulates the observed behavior of the pappus actuator.": It is necessary to show that the recapitulate is qualitative rather than quantitative.

We have clarified the sentence prior to this to demonstrate the similarities between the model in both quantitative and qualitative manners (line 396-8). Although our model does not perfectly match the experimental data quantitatively, our sensitivity analysis shows that the behavior is robust and we have a degree of motion that would be sufficient to make a substantial difference to pappus functionality during flight, as we mentioned in the discussion (line 623-4).

The above (line 396-8) mean (line 399-401)? Also, does (line 623-4) mean (line 627-630)? The line number seems to be off in the revision version. Please confirm. The line number of the correction or addition part is incorrect, but there is no objection to the content of the correction or addition.

19) P.22, L.544 "A 3D computational model may change this requirement as the remaining tissue would be compete connected.": It is necessary to mention the axisymmetric computational model before the 3D computational model. In the axisymmetric model, it (For example, Chien Wei- Zang, Finite element analysis of axisymmetric elastic body problems, Applied Mathematics and Mechanics volume 1, pages 23- 34 (1980)). In your paper, the analysis is performed assuming the expansion and contraction of the apical plate as a two-dimensional problem, but it is more appropriate to treat it

as an axisymmetric problem. When considered as an axisymmetric model, the volume of the outer cortex and side region is larger than that of the two-dimensional model, so it is expected that the strain and stress generated by swelling will also be larger. Therefore, it is considered to carry out the axisymmetric analysis is required in the next research step, but in this paper, it is necessary to consider whether it is more consistent with the experiment when considered as an axisymmetric model.

(Figure to show the contraction deformation of the apical plate model in the 2D plane strain condition and the 3D axisymmetric condition).

The red mesh is the deformed (dry) state given by the 2D reference model presented in and finally, the blue mesh is the deformed (dry) state given as the result of the 3D axisymmetric model where no boundary condition is applied, and all parameters (geometrical as well as those related to material properties) are taken the same as in the 2D reference model of the article. Although we find that indeed, the side region has higher area change in the axisymmetric case than in the 2D case, we also find that the difference between the two deformed geometries and in particular in the predicted side angle values is very small (see table below).

All area changes measurements presented in the article were done in 2D, on physically cut sections of the actuator tissue, and the difference in outcome between the 2D and the axisymmetric 3D model turns out to be below measurement errors. Therefore, we think that the 2D model is more appropriate to the experimental results presented in the article.

(Table to show the Area change (Ahydrated /Adry) and holding angle in 2D and axisymmetric 3D condition)

I respect you for performing a new 3D axisymmetric analysis and comparing it to the 2D plane strain analysis. As a result, it was found that the holding angle value hardly changed, and the effectiveness of the approximation by the plane strain condition in this study was confirmed. It is probable that the approximation of the plane strain condition was effective for this problem because the area (volume) of the part away from the central axis is large and a simple deformation of only contraction is imposed. I agree that the 3D model is a topic for the future, and only the analysis results of the 2D model is described in this paper. On the other hand, I think it would be good to describe in the paper that the plane strain condition is effective approximation for this problem and the reason for its effectiveness.

20) P.23, L.573, "we find from our computational model that intrinsic anisotropy is not necessary for a functional model. ": Since vasculature has a large anisotropy with respect to the fiber direction, it is expected that the influence on the expansion / contraction deformation of the entire apical plate is not small. Is it possible that the tendency shown in Figures S5 and S6 will change when the intrinsic anisotropy in the material is taken into account?

Thank you for this interesting point. We agree that anisotropy of some cell types is quite likely and would probably have an effect on the deformation. If intrinsic anisotropy is present, we would expect it to enhance the deformation further so our results represent a lower bound. On the magnitude of angle changes. Our results show that anisotropy is not inherently required for the morphing behavior we observe as we explain in the discussion (line 615-21). While additional complexity may be present in the tissue, it is not required for Uncovering these additional subtleties will be the focus of our future investigations.

(Line 615-21) is (line 621-27), isn't it? "Simply juxtaposing tissue types with differential isotropic swelling capacities and stiffnesses that arise solely from cell density recapitulates the behavior of the actuator (Fig 5)." There is no arguing about the above points, but can you explain in the paper why the approximation of isotropic contraction is effective because the change in holding angle is mainly caused by the differential swelling properties of each region?

These are my additional questions and comments. I hope this will help improve the quality of the paper.
exist?